# Genome-wide alterations of uracil distribution patterns in human DNA upon chemotherapeutic treatments

Hajnalka L Pálinkás[1,2,3]*, Angéla Békési[1,2], Gergely Róna[2,4,5,6], Lőrinc Pongor[7,8], Gábor Papp[2], Gergely Tihanyi[1,2], Eszter Holub[2], Adám Póti[9], Carolina Gemma[10], Simak Ali[10], Michael J Morten[4], Eli Rothenberg[4], Michele Pagano[4,5,6], Dávid Szűts[9], Balázs Győrffy[7,8], Beáta G Vértessy[1,2]*

[1]Genome Metabolism Research Group, Institute of Enzymology, Research Centre for Natural Sciences, Budapest, Hungary; [2]Department of Applied Biotechnology and Food Sciences, Budapest University of Technology and Economics, Budapest, Hungary; [3]Doctoral School of Multidisciplinary Medical Science, University of Szeged, Szeged, Hungary; [4]Department of Biochemistry and Molecular Pharmacology, New York University School of Medicine, New York, United States; [5]Perlmutter Cancer Center, New York University School of Medicine, New York, United States; [6]Howard Hughes Medical Institute, New York University School of Medicine, New York, United States; [7]Cancer Biomarker Research Group, Institute of Enzymology, Research Centre for Natural Sciences, Budapest, Hungary; [8]Department of Bioinformatics and 2nd Department of Pediatrics, Semmelweis University, Budapest, Hungary; [9]Genome Stability Research Group, Institute of Enzymology, Research Centre for Natural Sciences, Budapest, Hungary; [10]Department of Surgery and Cancer, Imperial College London, Hammersmith Hospital Campus, London, United Kingdom

*For correspondence:
palinkas.hajnalka@ttk.hu (HLP);
vertessy@mail.bme.hu (BGV)

Competing interests: The authors declare that no competing interests exist.

**Abstract** Numerous anti-cancer drugs perturb thymidylate biosynthesis and lead to genomic uracil incorporation contributing to their antiproliferative effect. Still, it is not yet characterized if uracil incorporations have any positional preference. Here, we aimed to uncover genome-wide alterations in uracil pattern upon drug treatments in human cancer cell line models derived from HCT116. We developed a straightforward U-DNA sequencing method (U-DNA-Seq) that was combined with in situ super-resolution imaging. Using a novel robust analysis pipeline, we found broad regions with elevated probability of uracil occurrence both in treated and non-treated cells. Correlation with chromatin markers and other genomic features shows that non-treated cells possess uracil in the late replicating constitutive heterochromatic regions, while drug treatment induced a shift of incorporated uracil towards segments that are normally more active/functional. Data were corroborated by colocalization studies via dSTORM microscopy. This approach can be applied to study the dynamic spatio-temporal nature of genomic uracil.

## Introduction

The thymine analogue uracil is one of the most frequent non-canonical bases in DNA appearing either by thymine replacing misincorporation or as a product of spontaneous or enzymatic cytosine deamination reaction (*Krokan et al., 2002*). Consequently, uracil in DNA is usually recognized as an error that is efficiently repaired by the multistep base excision repair (BER) pathway initiated by uracil-DNA glycosylases (UDGs) (*Krokan and Bjørås, 2013*; *Wallace, 2014*). In other respects, uracil in

DNA is known to be involved in several physiological processes (e.g. antibody maturation [*Liu and Schatz, 2009*; *Maul and Gearhart, 2010*; *Maul and Gearhart, 2014*; *Xu et al., 2012*], antiviral response [*Burns et al., 2015*; *Stenglein et al., 2010*], insect development [*Horváth et al., 2013*; *Muha et al., 2012*]), however, the exact mechanism and regulation of uracil-DNA metabolism including the roles of UDGs need to be elucidated. There are four known members of the UDG family in humans: (i) the most active uracil-DNA glycosylase encoded by the *ung* gene (UNG1 mitochondrial and UNG2 nuclear isoform), (ii) the single-strand selective monofunctional uracil-DNA glycosylase 1 (SMUG1), (iii) thymine DNA glycosylase (TDG specialized for repair of T:G and U:G) and (iv) methyl CpG binding domain protein 4 (MBD4 repairs U:G) (*Visnes et al., 2009*). UNG2 removes most of the genomic uracil from both single- and double-stranded DNA regardless of the uracil originating from mutagenic cytosine deamination or thymine replacing misincorporation (*Kavli et al., 2002*).

Thymine replacing uracil misincorporation is normally prevented by the tight regulation of the cellular dUTP/dTTP ratio maintained by two enzymes, the dUTPase and the thymidylate synthase. The dUTPase enzyme removes dUTP from the cellular pool by catalyzing dUTP hydrolysis into dUMP and $PP_i$ (*Vértessy and Tóth, 2009*). Lack or inhibition of dUTPase leads to increased dUTP levels and under such conditions, DNA polymerases readily incorporate uracil opposite to adenine. Similarly, several anticancer drugs (such as 5-fluorouracil (5-FU), 5-fluoro-2'-deoxyuridine (5FdUR), capecitabine, methotrexate, raltitrexed (RTX), pemetrexed) target the de novo thymidylate synthesis pathway *via* thymidylate synthase inhibition to perturb the tightly regulated dUTP/dTTP ratio, eventually triggering thymineless cell death (*Blackledge, 1998*; *Requena et al., 2016*; *Wilson et al., 2014*). Although the exact molecular mechanism is not yet fully understood, massive uracil misincorporation, hyperactivity of the repair process and/or stalling of the replication fork are all suggested to be involved in the process (*Khodursky et al., 2015*; *Ostrer et al., 2015*). UNG has been suggested to play a key role in this mechanism, as being responsible for the initiating step in uracil removal that may lead to futile cycles if the cellular dUTP/dTTP ratio is elevated. A quantitative insight into the magnitude and the pattern of uracil incorporation into genomic DNA as induced by these chemotherapeutic treatments is expected to contribute to a better understanding of the cell death mechanism induced by the respective drugs.

Direct observation of the uracil moieties incorporated upon drug treatments have been hampered by the efficient and fast action of UNG. To overcome this problem, we wished to counteract the action of UNG in human cells *via* introduction of the well characterized, specific UNG inhibitor, UGI (*Luo et al., 2008*; *Mol et al., 1995*) into the cellular milieu. It has already been shown that UGI expression does not affect either the cytotoxicity, or the DNA damage and cell cycle response upon RTX and 5FdUR treatment (*Luo et al., 2008*). Using UGI expressing cell lines, we aimed to reveal the nascent pattern of uracil moieties in DNA induced by perturbation of thymidylate metabolism both using genome-wide uracil-specific sequencing and in situ cellular imaging of uracils within human genomic DNA. Previously, we designed a uracil-DNA (U-DNA) sensor tailored from an inactive mutant of human UNG2 that was successfully applied in semi-quantitative dot blot analysis and direct immunocytochemistry (*Róna et al., 2016*). Some additional approaches have also been published to detect uracil-DNA within its genomic context such as (i) techniques focusing on specific, well-defined regions of the genome (qPCR [*Horváth and Vértessy, 2010*] and 3D-PCR [*Suspène et al., 2005*]), (ii) techniques that have been applied only to smaller sized genomes (Excision-seq [*Bryan et al., 2014*] and UPD-seq [*Sakhtemani et al., 2019*]), and (iii) techniques requiring labor-intensive isolation and multistep processing of genomic DNA samples (dU-seq [*Shu et al., 2018*]).

Here, we employ the U-DNA sensor in a DNA-IP-seq-like (DIP-seq-like) approach (termed as U-DNA-Seq) and develop a robust bioinformatic pipeline specifically designed for reliable interpretation of next generation sequencing (NGS) data for genome-wide distribution of uracil. We selected two drugs, RTX (raltitrexed, or tomudex) and 5FdUR that perturb thymidylate biosynthesis with different modes of action and analyzed their effects on genomic uracil distribution. These two drugs are frequently applied in treatment of colon cancers, therefore we chose a human colon carcinoma cell line, HCT116 and its mismatch repair (MMR) proficient variant as well-established and relevant cellular models (*Koi et al., 1994*; *Meyers et al., 2001*; *Rashid et al., 2019*). We show that drug treatment led to increased probability of uracil incorporation into more active chromatin regions in HCT116 cells expressing the UNG inhibitor protein UGI. In contrast, uracil was rather restricted to constitutive heterochromatic regions both in wild type cells and in non-treated UGI-expressing cells.

Moreover, we further developed the U-DNA sensor-based staining method (*Róna et al., 2016*) that now uniquely allows in situ microscopic visualization of uracil in human genomic DNA. Confocal and super-resolution microscopy images and colocalization measurements strengthened the sequencing-based distribution patterns.

## Results

### Genome-wide mapping of uracil-DNA distribution patterns by U-DNA-Seq

We designed an adequate DNA immunoprecipitation method that can provide U-DNA specific genomic information by NGS. This method, termed U-DNA-Seq is based on the rationale of the well-established DIP-seq technology. *Figure 1A* presents the scheme of the protocol leading to an enriched U-DNA sample that was then subjected to NGS. Immunoprecipitation was carried out by applying the FLAG-tagged catalytically inactive ΔUNG sensor (described in *Róna et al., 2016*) to bind to uracil in purified and fragmented genomic DNA, followed by a pull-down with anti-FLAG agarose beads. All samples addressed by the U-DNA-Seq in the present study are summarized in *Supplementary file 1*-table 1.

To allow better detection of nascent uracil, the UNG-inhibitor UGI was expressed in both MMR deficient and proficient HCT116 cells to prevent the action of the major uracil-DNA glycosylase. Besides transient transfection, stable UGI-expressing HCT116 cell lines were also established by retroviral transduction of human codon optimized UGI along with EGFP (*Figure 1—figure supplement 1A*). We proceeded to treat the UGI-expressing cells with either 5FdUR or RTX. Notably, this combination of UGI expression and drug treatment did not result in any observable cell death. As shown in *Figure 1—figure supplement 1B–D*, UGI expression and drug (5FdUR or RTX) treatment led to significantly increased uracil content in genomic DNA that is even more pronounced in case of the MMR proficient cells. It is important to note that either UGI expression or treatments with drugs targeting de novo thymidylate biosynthesis pathways on their own do not lead to elevated U-DNA level (*Luo et al., 2008*; *Róna et al., 2016*; *Yan et al., 2016*). Following U-DNA immunoprecipitation, successful enrichment of U-DNA could be confirmed by dot blot assay in the case of drug-treated cells (5FdUR_UGI or RTX_UGI, *Figure 1B*). To further confirm the capability of U-DNA-IP, uracil-containing spike-in DNA was combined with non-treated genomic DNA samples (Materials and methods). In these samples U-DNA-IP led to 4.5 fold enrichment of the uracil-containing spike-in DNA compared to the uracil-free spike-in as determined by qPCR. Specificity of U-DNA immunoprecipitation is also underlined by the fact that pull-down with empty anti-FLAG beads not containing the U-DNA sensor (i.e. negative control) resulted in negligible amount of DNA (less than 5%, *Figure 1—figure supplement 2A*, see also *Supplementary file 1*-table 1). Still, genome-wide sequencing data could be obtained from these negative control samples as well. We demonstrated that subtracting such control signals (for details see *Supplementary file 1*) will not affect the detected uracil distribution pattern regardless if the sample was drug-treated or not (*Figure 1—figure supplement 2B–C*). These control experiments provided confidence about the applicability and specificity of our U-DNA-IP method.

Then, enriched and input DNA samples both from treated (5FdUR_UGI, 5FdUR_UGI_MMR, RTX_UGI, and RTX_UGI_MMR) and non-treated (wild type (WT), NT_UGI, and NT_UGI_MMR) samples were subjected to library preparation and NGS. U-DNA-Seq was carried out in two independent biological replicates for each sample. We also performed U-DNA-Seq on non-treated wild type K562 cells in order to have a reference point to the published dU-seq data (*Shu et al., 2018*).

Sequencing data were analyzed using the herein developed computational pipeline shown in *Figure 2* (for more details see the *Figure 2—figure supplement 1* and *Supplementary file 1*-table 2). When reads were aligned to the reference GRCh38 human genome, only uniquely mapped reads were kept and regions suffering from alignment artefacts were excluded from the analysis by blacklisting (*Figure 2—figure supplement 2*). Statistics on pre-processing steps are shown in *Supplementary file 1*-table 3. Correlation among the samples at the level of cleaned aligned reads (bam files) was checked by Pearson correlation analysis (*Figure 2—figure supplement 3*, for details see *Supplementary file 1*). Here, a clear difference was found between the input and the enriched

samples; input samples were more similar to each other regardless the applied treatment, while the enriched drug-treated and non-treated samples showed dramatic differences.

There are two principal approaches to extract the signals of uracil enrichment from the cleaned aligned reads: (1) computing genome scaled coverage and log2 ratio tracks, and (2) peak calling that is conventionally used for ChIP-seq data analysis. Log2 ratio tracks provide more detailed information on the uracil-DNA distribution patterns, however, it is not compatible with efficient screening on large dataset (*Figure 2* and *Figure 2—figure supplement 1*). Hence, we generated interval (bed) files from the log2 ratio tracks for each sample (*Figure 3A*) that contain simplified information on uracil enriched regions as described in the *Supplementary file 1*. Then, we evaluated both the regions derived from the log2 ratio tracks, and the peak calling results (*Figure 3—figure supplement 1* and *Figure 3—figure supplement 2*). We found that the uracil enriched genomic regions are rather broad and much less intense than conventional peaks in ChIP-seq for transcription factors or even for histone modifications. This is somehow expected considering basically stochastic nature of uracil occurrence *via* both misincorporation and spontaneous cytosine deamination. In agreement with this, reliability and reproducibility of the peak calling approach (using MACS2 with 'broad'

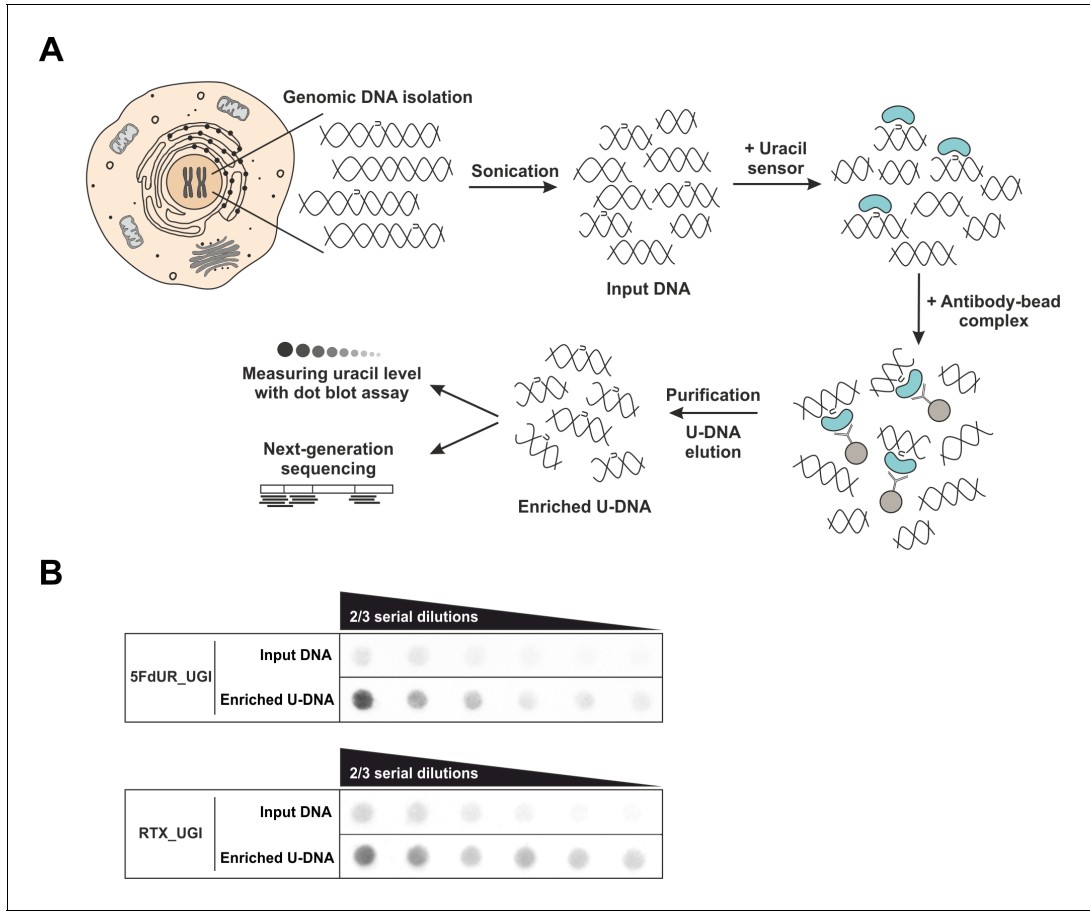

**Figure 1.** U-DNA-Seq provides genome-wide mapping of uracil-DNA distribution. (**A**) Schematic image of the novel U-DNA immunoprecipitation and sequencing method (U-DNA-Seq). After sonication, enrichment of the fragmented U-DNA was carried out by the 1xFLAG-ΔUNG sensor construct followed by pull-down with anti-FLAG agarose beads. U-DNA enrichment compared to input DNA was confirmed by dot blot assay before samples were subjected to NGS. (**B**) Immunoprecipitation led to elevated uracil levels in enriched U-DNA samples compared to input DNA in case of both 5FdUR (5FdUR_UGI) and RTX (RTX_UGI) treated, UGI-expressing HCT116 samples. For each treatment, the same amount of DNA was loaded as input and enriched U-DNA samples providing a correct visual comparison. Two-third serial dilutions were applied.

The online version of this article includes the following source data and figure supplement(s) for figure 1:

**Figure supplement 1.** Elevation of genomic uracil content upon stable UGI expression and drug treatments.

**Figure supplement 1—source data 1.** Quantification of genomic uracil content based on densitometry of dot blot measurements.

**Figure supplement 2.** Negative control of U-DNA-IP using ΔUNG sensor free (empty) anti-FLAG beads.

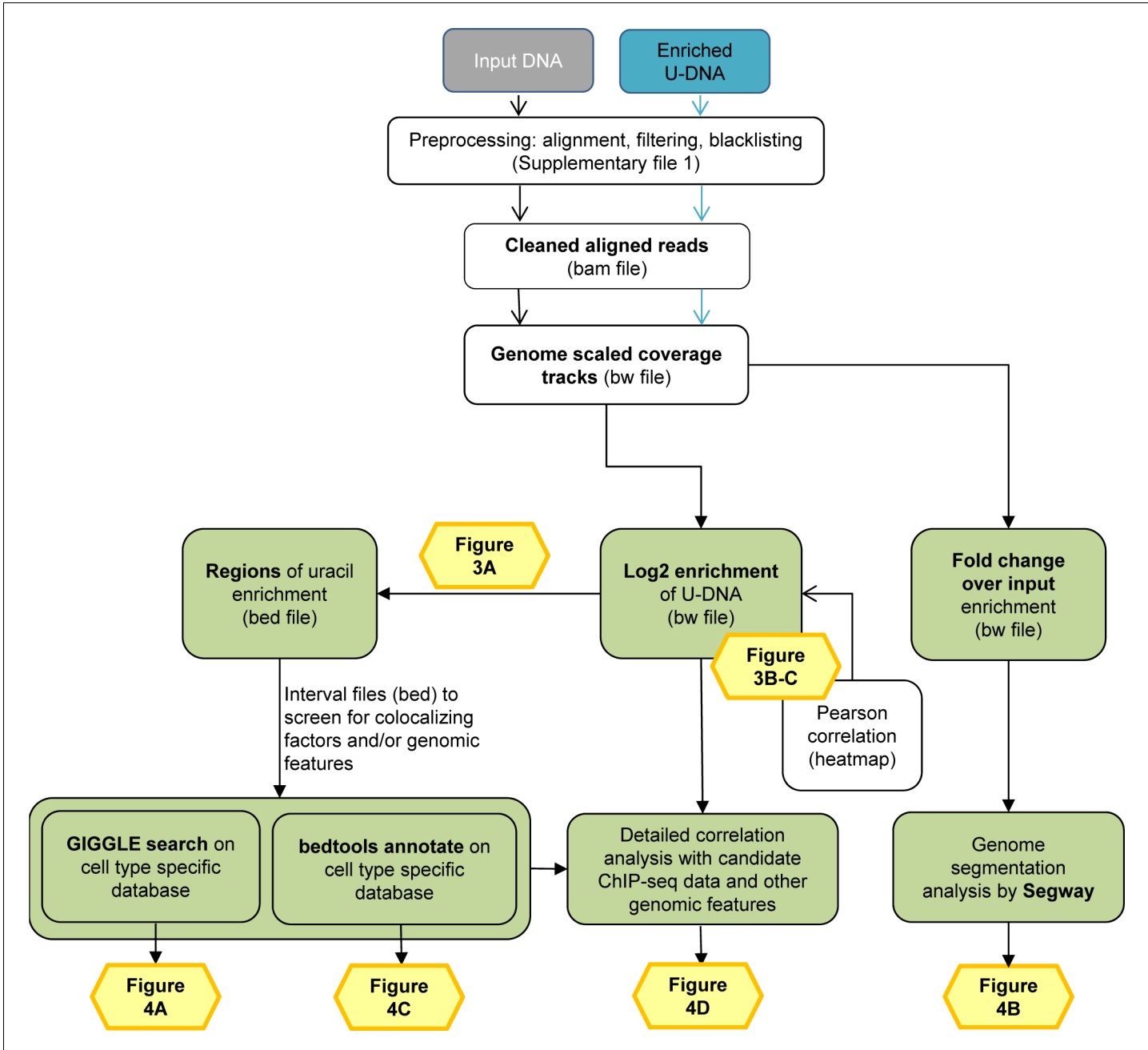

**Figure 2.** Data analysis pipeline. Both input and enriched U-DNA samples were pre-processed the same way: initial trimming and alignment were followed by filtering for uniquely mapped reads and blacklisting of regions suffering from alignment artefacts, resulting in cleaned read alignments in the format of bam files. The key steps of our proposed data processing are (1) calculation of genome scaled coverage tracks (bigwig/bw files), (2) calculation of log2 (enriched coverage/input coverage) ratio tracks (bigwig/bw files), (3) extraction of interval (bed) files of uracil enriched regions from the corresponding log2 ratio tracks. To correlate the uracil enrichment profiles with other published data, first quick screens using interval files were done, and then detailed correlation analysis with a promising candidate of colocalizing genomic features was performed using coverage track files. GIGGLE search (*Layer et al., 2018*) and bedtools annotate (*Quinlan and Hall, 2010*) were used for scoring the similarities between query uracil-DNA and the database interval files. Genome segmentation analysis was performed on fold change over input bigwig files either from the ENCODE database, or our own ChIP-seq data and U-DNA profiles using Segway package (*Chan et al., 2018*; *Hoffman et al., 2012*). Figures corresponding to the different analysis steps are also indicated. A more detailed pipeline is shown in *Figure 2—figure supplement 1*, and the full methodology is described in the *Supplementary file 1*, *3–5*.

The online version of this article includes the following source data and figure supplement(s) for figure 2:

**Figure supplement 1.** Detailed analysis pipeline.
**Figure supplement 2.** Creation of cell line specific blacklist.

*Figure 2 continued on next page*

*Figure 2 continued*

**Figure supplement 2—source data 1.** Histograms to determine the UHS and the low mappability regions for cell line specific blacklists.
**Figure supplement 3.** Pearson correlation among processed BAM files.

option) was found to be clearly suboptimal for determination of uracil distribution patterns (*Figure 3—figure supplement 1* and *Figure 3—figure supplement 2*). Therefore, we decided to proceed with the coverage track approach rather than the peak calling. All of the main figures rely on analysis performed with either the log2 ratio tracks or the regions of uracil enrichment derived from the log2 ratio tracks.

*Figure 3A* shows the uracil distribution pattern in a selected chromosomal segment where an uneven distribution with variably spaced broad regions is observed (the same data for all the chromosomes are shown in *Supplementary file 2*). A clear difference between non-treated and drug-treated cells is already obvious from this view, and the correlations were also measured quantitatively on the whole log2 ratio tracks by Pearson correlation coefficients and related scatter plots (*Figure 3B*, for description of the samples see *Supplementary file 1*-table 1, for individual replicates see *Figure 3—figure supplement 3*). Interestingly, the impact of MMR proficiency on the uracil distribution pattern is obvious in case of the 5FdUR treatment, while RTX treated and especially the non-treated samples do not show notable differences compared to their MMR deficient counterparts.

The uracil-enrichment coverage tracks in *Figure 3A* and the related correlations in *Figure 3B* already revealed altered distribution of uracil-containing regions in the drug-treated as compared to the non-treated samples. This difference was further underlined in a histogram representation of uracil enrichment signal (*Figure 3C* and *Figure 3—figure supplement 4*) where drug treatment led to a higher number of genomic segments (more data bins) with increased uracil level. MMR proficiency in case of the 5FdUR treatment substantially changed this phenomenon. We investigated whether the uracil distribution patterns might show correlation to any previously determined genomic features. For this reason, we built a relevant database by collecting cell type specific ChIP-seq and DNA accessibility data (for details see *Supplementary file 3–4*).

Interrogation of the constructed specialized database with respect to the uracil-DNA distribution patterns was performed using interval (bed) files of uracil enriched regions (derived from log2 ratio track) for each U-DNA-Seq sample. To screen for similarity between sample and database interval (bed) files, we applied the GIGGLE search tool (for details see *Supplementary file 3*). GIGGLE scores measure the colocalization independently from the size of the compared intervals (*Layer et al., 2018*). Each interval file in the database corresponded to a ChIP-seq data with a given factor (e.g. histone markers, transcription factors, etc.). GIGGLE scores were then calculated pairwise (each sample to each database interval file), and plotted for the top ten factors corresponding to the highest scores (*Figure 4A*, full data are presented in *Supplementary file 3*-table 1). The similarity scores of the U-DNA-Seq data with regard to the different chromatin markers indicate that non-treated cells may possess uracils preferentially in the constitutive heterochromatin (high scores with H3K9me2 and H3K9me3 [*Hyun et al., 2017*; *Saksouk et al., 2015*]). On the other hand, drug treatment of the cells either with 5FdUR or RTX, induces uracil incorporation into more active genomic segments, which correlates with euchromatin histone marks (H3K36me3 [*Becker et al., 2017*; *Hyun et al., 2017*; *Pfister et al., 2014*], H3K4me1/3 [*Hyun et al., 2017*], H3K27ac [*Creyghton et al., 2010*], H3K9ac [*Gates et al., 2017*]), or factors associated to either activation or repression in a context dependent manner (SP1 [*Doetzlhofer et al., 1999*], H3K27me3 [*Becker et al., 2017*; *Saksouk et al., 2015*], H2AZ/AFZ [*Giaimo et al., 2019*]) (*Figure 4A*). Interestingly, MMR proficiency has an impact on this correlation in case of both drug-treated samples reflecting in decreased GIGGLE scores.

In order to decide whether drug treatments may cause any notable changes in the distribution pattern of epigenetic markers as compared to the normal patterns, we have performed a direct comparative ChIP-seq study on our UGI-expressing HCT116 cell line. For this, we have selected the H3K36me3 histone marker that gave the highest GIGGLE scores with the RTX treated U-DNA pattern. ChIP-seq for H3K36me3 was performed in both non-treated (NT_UGI_H3K36me3), and RTX treated (RTX_UGI_H3K36me3) UGI-expressing HCT116 cells (Materials and methods, for the details

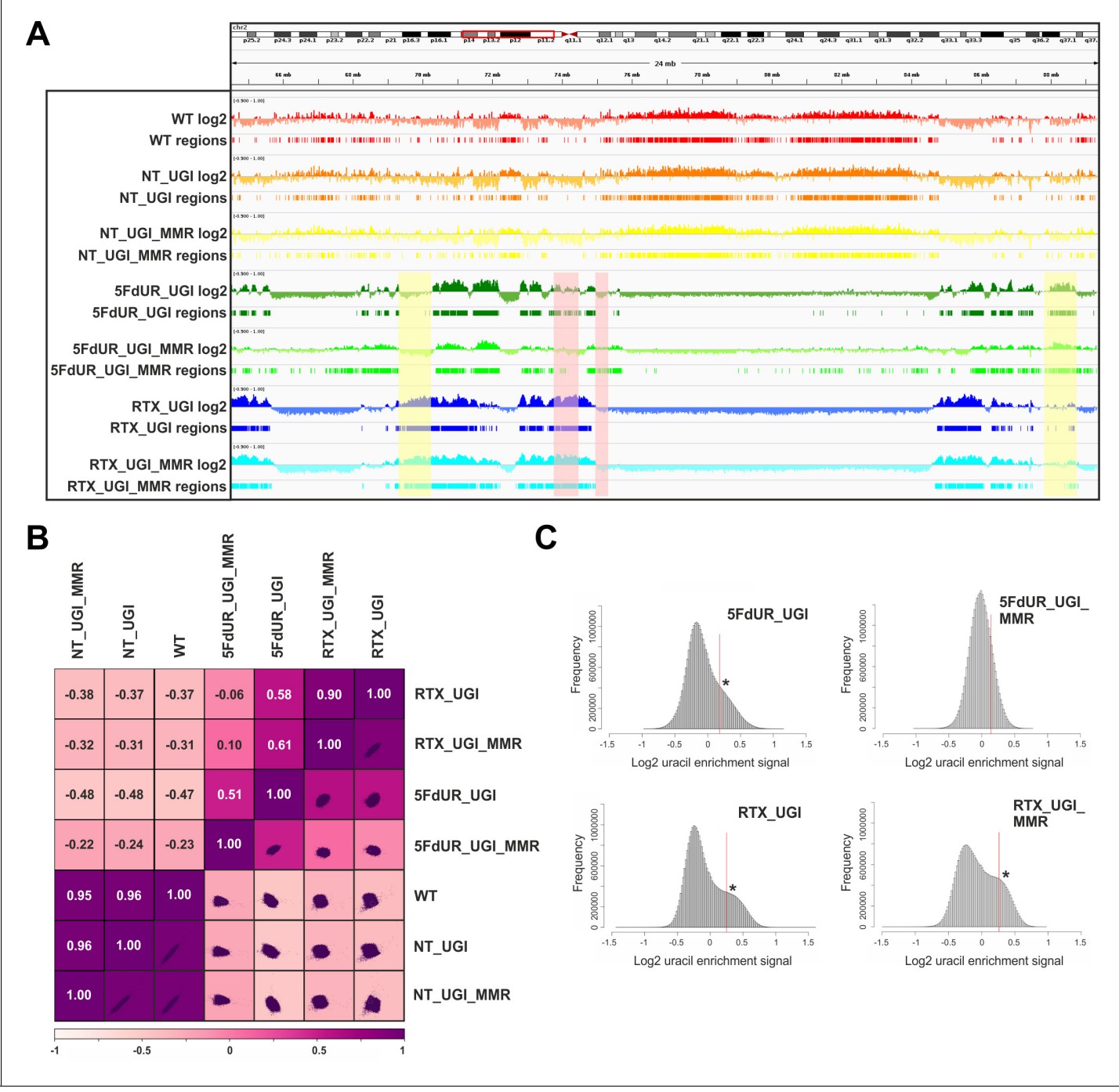

**Figure 3.** Comparison of processed U-DNA-Seq data among samples. (**A**) Representative IGV view in genomic segment (chr2:64,500,000–89,500,001) shows log2 ratio signal tracks of enriched versus input coverage (log2, upper tracks) and derived regions of uracil enrichment (regions, bottom tracks) for non-treated: wild type (WT, red), UGI-expressing (NT_UGI, orange), and MMR proficient UGI-expressing (NT_UGI_MMR, yellow); and for treated: with 5FdUR (5FdUR_UGI, green; 5FdUR_UGI_MMR, light green) or raltitrexed (RTX_UGI, blue; RTX_UGI_MMR, cyan) HCT116 samples. Two replicates for each sample were merged before coverage calculation. Differences between treated and non-treated samples are clearly visible. Furthermore, 5FdUR and RTX treatments caused similar but not identical uracil enrichment profiles (differences are highlighted with yellow shade). The impact of the MMR status in case of the 5FdUR treated samples is highlighted with pink shade. (**B**) Comparison of log2 uracil enrichment profiles among samples was performed using multiBigwigSummary (deepTools) and Pearson correlation was plotted using plotCorrelation (deepTools). A heatmap combined with scatter plots is shown for the seven samples. Two replicates for each sample were merged before coverage calculation, and the same analysis for individual replicates are shown in *Figure 3—figure supplement 3*. (**C**) Histograms of log2 ratio profiles were calculated and plotted using R for the drug-treated samples. A sub-population of data bins with elevated log2 uracil enrichment signal is clearly visible (indicated with asterisk) in most cases,

*Figure 3 continued on next page*

*Figure 3 continued*

where high uracil incorporation was detected (*Figure 1—figure supplement 1B–D*). Thresholds applied in determination of uracil enriched regions are indicated with red line and also provided in *Figure 3—source data 1* together with the histogram data.

The online version of this article includes the following source data and figure supplement(s) for figure 3:

**Source data 1.** Histograms for the U-DNA signal distribution in drug-treated samples.
**Figure supplement 1.** Representative IGV views from aligned reads to a 10 Mb cluster of uracil enriched regions compared to usual peak calling results.
**Figure supplement 2.** Comparison of the regions of uracil enrichment derived from log2 signal ratio and the peaks called by MACS2.
**Figure supplement 3.** Pearson correlation among log2 ratio tracks of replicates.
**Figure supplement 4.** Histograms for the U-DNA signal distribution in non-treated samples.
**Figure supplement 4—source data 1.** Histograms for the U-DNA signal distribution in non-treated samples.

of the analysis see *Supplementary file 3*, for description of the samples see *Supplementary file 1*-table 1). Comparison of our H3K36me3 ChIP-seq data to those available within the ENCODE database is presented in *Figure 4—figure supplement 1*. These results reveal no substantial differences between RTX_UGI_H3K36me3 and NT_UGI_H3K36me3 samples suggesting no major chromatin rearrangement upon drug treatment. Moreover, our ChIP-seq data are similar to the corresponding ENCODE data. Furthermore, on *Figure 4A*, GIGGLE scores between the U-DNA patterns and our own ChIP-seq peaks are also indicated and these are in good agreement with the other corresponding scores in case of both the non-treated and RTX treated samples.

To understand broader genome-wide correlations, a genome segmentation approach was employed using Segway software (see details in *Supplementary file 3*). 22 independent, HCT116 related ChIP-seq experiments of the ENCODE database were selected for the analysis together with our U-DNA-Seq data and also our ChIP-seq data for H3K36me3 (NT_UGI_H3K36me3 and RTX_U-GI_H3K36me3). 25 genomic segments were defined and identified with the signal distribution presented in *Figure 4B*. This analysis on one hand confirmed the correlations that had already been suggested by the GIGGLE search; on the other hand revealed that the histone markers are not the most correlating genomic features. The drug treatment induced shift towards the transcriptionally more active regions is also reflected in the segments 14 and 21, where treated samples show slightly increased U-DNA signal, in contrast to the definitely low signal in case of the non-treated samples. Moreover, it was also confirmed that the most correlating histone markers are the H3K36me3 and the H3K27me3 for the U-DNA pattern of the RTX (segments 19 and 8) and the 5FdUR (segments 17 and 4) treated samples, respectively. The differences between the two drug treatments (e.g. regarding the histone markers mentioned above), and also between the corresponding MMR deficient and proficient cells (e.g. MMR dependent decreased signal intensities in the segments (21 and 14) associated with active transcription), seem to be coherent with the GIGGLE analysis. Similarly, in the case of the non-treated samples, the H3K9me3 constitutive heterochromatin marker was confirmed to be the most correlating histone marker (segments 1 and 24). Nevertheless, the highest U-DNA signal segments are not matching with any of the investigated histone markers (see segments 18 for the non-treated; and 0 and 15 for drug-treated samples). The fact that the histone markers are not the most correlating genomic features prompted us to further search for potential correlating features.

Therefore, we investigated colocalization of U-DNA enriched regions with different coding properties, CpG islands, active regions based on DNase hypersensitivity, different types of repetitive segments, giemsa stained cytogenetic bands and different replication timing. Bedtools annotate software (*Quinlan and Hall, 2010*) was used to extract the number of overlapping bases. Scores measuring the colocalization are presented in *Figure 4C* for a systematic selection of the tested features. The results of the full analysis are provided in *Supplementary file 4*-table 1. The data suggest that uracil incorporation in transcriptionally active (e.g. active promoters, DNase hypersensitive sites) and potentially active genomic segments (CpG islands, genes, especially exons and CDS regions), is increased upon drug treatment, both in MMR deficient and proficient cells, although to different extents. The proposed uracil enrichment in transcriptionally active genomic regions is also in agreement with the colocalization with different repeat classes: the drug-treated samples show higher colocalization with short interspersed nuclear elements (SINEs [*Kramerov and Vassetzky, 2005*]) and long terminal repeats (LTRs [*Kovalskaya et al., 2006*]) which are known to be more frequently transcribed as compared to long interspersed nuclear elements (LINEs [*Boissinot and Furano,*

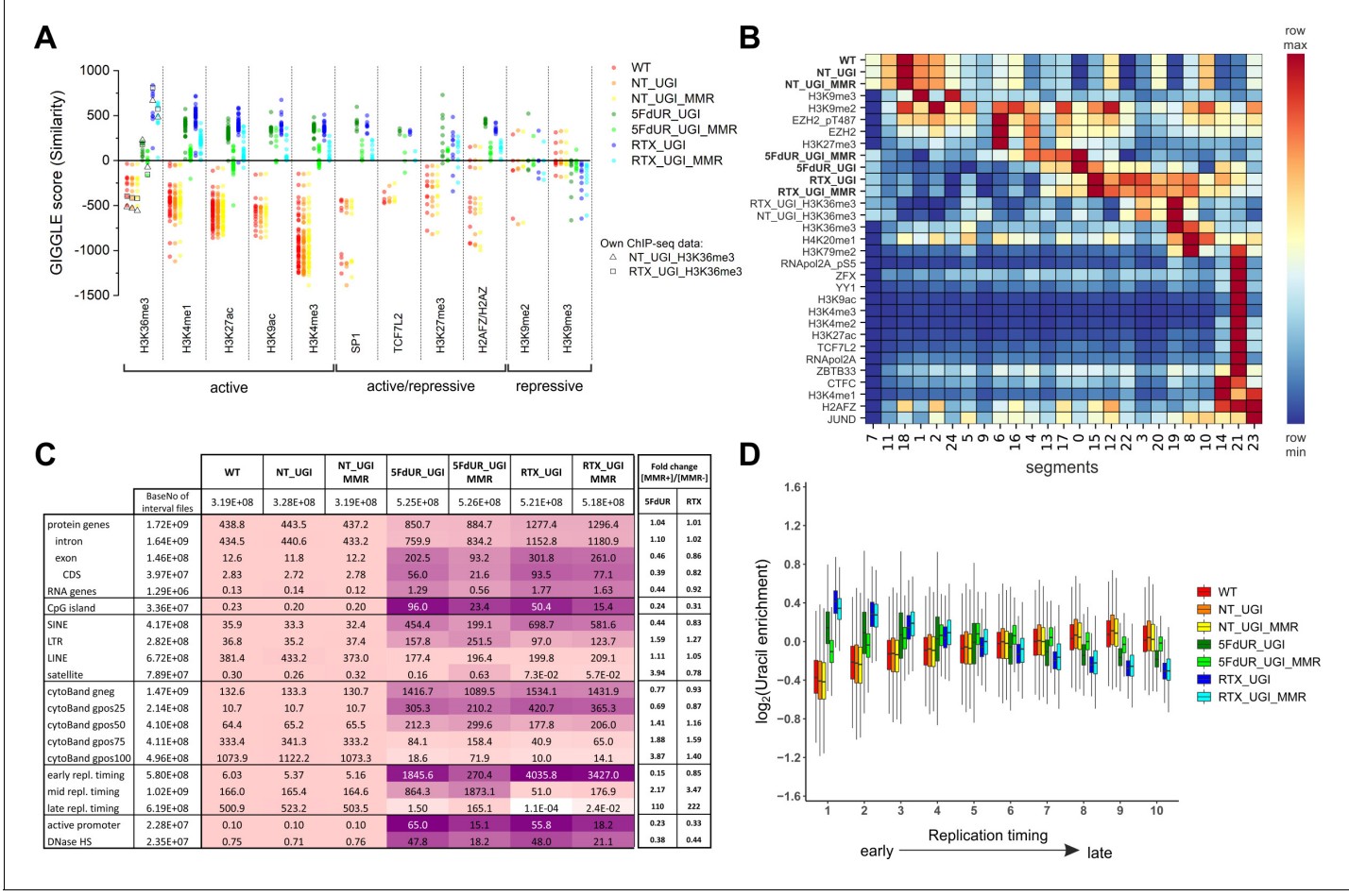

**Figure 4.** Characterization of U-DNA enrichment patterns. (**A**) GIGGLE search was performed with interval (bed) files of uracil enriched regions on a set of HCT116 related ChIP-seq and DIP-seq experimental data (for details see ***Supplementary file 3***). Factors corresponding to the top 10 hits for each sample were selected. GIGGLE scores between all seven samples and all experiments corresponding to these factors were plotted excluding those, where data were not informative (data are found in ***Supplementary file 3***-table 1). Source data are available in ***Figure 4—source data 1***. Histone marks and the transcription factors, SP1 and TCF7L2 are categorized depending on their occurrence in transcriptionally active or repressive regions. Notably, some of them have plastic behavior allowing either transcriptionally active or repressive function. U-DNA-Seq samples are as follows: non-treated wild type (WT, red), non-treated UGI-expressing (NT_UGI, orange), 5FdUR treated UGI-expressing (5FdUR_UGI, green) and RTX treated UGI-expressing (RTX_UGI, blue) HCT116 cells, and their MMR proficient counterparts (NT_UGI_MMR, yellow; 5FdUR_UGI_MMR, light green; RTX_UGI_MMR, light blue). GIGGLE scores are also indicated for our own H3K36me3 ChIP-seq experiments (RTX_UGI sample: empty squares, NT_UGI sample: empty triangles). The tendencies are even more pronounced if the RTX treated U-DNA-Seq is compared with the RTX treated ChIP-seq or if the non-treated U-DNA-Seq is compared with the non-treated ChIP-seq data. (**B**) Genome segmentation analysis was performed on signal tracks of 22 ChIP-seq data available for HCT116 cells in the ENCODE database, on our own ChIP-seq data for H3K36me3, and on the seven U-DNA enrichment profiles (bold). The Segway train was performed with 25 labels and the corresponding genomic segments were identified with Segway annotate (***Chan et al., 2018***). The signal distribution data were calculated using Segtools (***Buske et al., 2011***), and plotted using python seaborn/matplotlib modules (***Hunter, 2007***). Source data are available in ***Figure 4—source data 2***. Details including the applied command lines are provided in ***Supplementary file 3***. The color-code is applied for each factor (rows) independently, from the minimum to the maximum value as indicated. (**C**) Correlation with genomic features. Interval (bed) files of genomic features were obtained from UCSC, Ensembl, and ReplicationDomain databases (for details see ***Supplementary file 4***-table 1), and correlation with interval files of uracil regions were analyzed using bedtools annotate software (details are provided in ***Supplementary file 4***). Numbers of overlapping base pairs were summarized for each pair of interval files, and scores were calculated according the formula: (baseNo_overlap/baseNo_sample_file) * (baseNo_overlap/baseNo_feature_file) * 10000. Heatmap was created based on fold increase of the scores compared to the corresponding WT scores. Sizes of interval files in number of base pairs are also given in the second column and the second line. Upon drug treatments, a clear shift from non-coding/heterochromatic/late replicated segments towards more active/coding/euchromatic/early replicated segments can be seen. CDS, coding sequence; SINE, short interspersed element; LTR, long terminal repeat; LINE, long interspersed element; cytoBand, cytogenetic chromosome band negatively (gneg) or positively (gpos) stained by Giemsa; repl. timing, replication timing; DNaseHS, DNase hypersensitive site. (**D**) Correlation analysis with replication timing. Replication timing data (bigWig files with 5000 bp binsize) specific for HCT116 were downloaded from ReplicationDomain database (***Weddington et al., 2008***). Data bins were distributed to 10 equal size

*Figure 4 continued on next page*

*Figure 4 continued*

groups according to replication timing from early to late. Then log2 uracil enrichment signals for these data bin groups were plotted for each sample using R (*Supplementary file 5*). Source data are available in *Figure 4—source data 3*.

The online version of this article includes the following source data and figure supplement(s) for figure 4:

**Source data 1.** GIGGLE similarity scores between U-DNA patterns and selected histone marks or transcription factors.
**Source data 2.** Signal distribution data from genome segmentation analysis by Segway.
**Source data 3.** Correlation between U-DNA patterns and replication timing.
**Figure supplement 1.** Comparison of our own H3K36me3 ChIP-seq data to each other and to the ENCODE data using Pearson correlation.
**Figure supplement 2.** IGV view of log2 ratio and regions of uracil enrichment on chromosome 1 (for all chromosomes see *Supplementary file 2*).
**Figure supplement 3.** Replication timing scores and AT content calculated on the genomic segments that were defined by the Segway analysis.
**Figure supplement 3—source data 1.** Replication timing scores and AT content calculated on genomic segments that were determined by the Segway analysis.

*2005*]) and satellite segments (*López-Flores and Garrido-Ramos, 2012*). It is interesting to note that MMR proficiency has an impact on this pattern also but only in case of the 5FdUR treatment.

The observed similarity between wild type uracil distribution and the patterns of histone markers associated with heterochromatin (*Figure 4A–B*) is further underlined by the positive correlation between U-DNA and cytogenetic chromosome G-bands (*Figure 4C*). Dark G-bands stained strongly by Giemsa were shown to correlate with AT-rich, heterochromatic, late replicating genomic segments (*Gilbert, 2002*; *Holmquist et al., 1982*). In contrast, negative G-bands are correlated better to the drug-treated uracil-DNA distribution pattern, also in agreement with our results from the comparison to histone markers (*Figure 4A–B*). Consistently, similar difference between patterns of U-DNA in non-treated versus drug-treated cells in early or late replicating genomic segments is also revealed. Late replicating regions are better correlated to the U-DNA distribution in non-treated cells, while the drug treatment induced U-DNA pattern is more similar to the early replicating segments (*Figure 4C*). Interestingly, in the 5FdUR treated samples, MMR proficiency led to a major decrease in the correlation between the U-DNA pattern and the early replicating segments, still the difference as compared to the non-treated samples remains. It is widely accepted that replication timing strongly correlates with chromatin structure, namely the open euchromatin and the condensed heterochromatin replicates in early and late S-phase, respectively (*Gilbert, 2002*). The correlation between U-DNA enrichment and replication timing was further analyzed using a better resolved time scale of replication (*Figure 4D*) which strengthened the initial observation. The correlations with G-banding and replication timing are also clearly visible on IGV views in *Figure 4—figure supplement 2*. Furthermore, colocalization with AT-rich heterochromatin for non-treated and GC-rich euchromatin for drug-treated samples is also reflected by the base composition of uracil enriched regions (*Figure 3—figure supplement 2A*). The surprisingly high correlation between uracil enrichment in drug-treated cells and CpG islands (*Figure 4C*) coincides with the elevated GC content of uracil enriched genomic regions in these samples. The replication timing correlation and the AT content were also calculated to the genomic segments identified by the Segway (cf. *Figure 4B*), and the above correlation was confirmed (*Figure 4—figure supplement 3*).

As the uracil distribution pattern in drug-treated cells shows correlation with the early replication timing, we wish to directly investigate if there is any cell cycle arrest occurring under our experimental conditions. *Figure 5* shows characteristic scatter plots indicating an expected cell cycle arrest in the drug-treated cells, namely delayed S-phase entry and progression (*Blackledge, 1998*; *Ding et al., 2019*; *Huehls et al., 2016*; *Yan et al., 2016*; *Zhao et al., 2016*). In agreement with the literature (*Luo et al., 2008*), our data clearly show no observable cell cycle effect of UGI expression in our non-treated samples (*Figure 5A*). Our data also revealed that the MMR proficiency somewhat tempers the observed cell cycle arrest, especially in case of the 5FdUR treatment (*Figure 5B*). As expected (*Meyers et al., 2001*), 5FdUR and RTX treatments eventually lead to DNA double-strand breaks (DSBs) as measured by yH2AX staining (*Figure 5—figure supplement 1*). DNA damage induction by the drugs was similar in MMR deficient and proficient HCT116 cells.

## In situ detection of U-DNA using super-resolution microscopy

We aimed to correlate genome-wide uracil distribution patterns in situ with chromatin architecture. Therefore, we further developed the U-DNA sensor constructs (*Róna et al., 2016*) to allow in situ

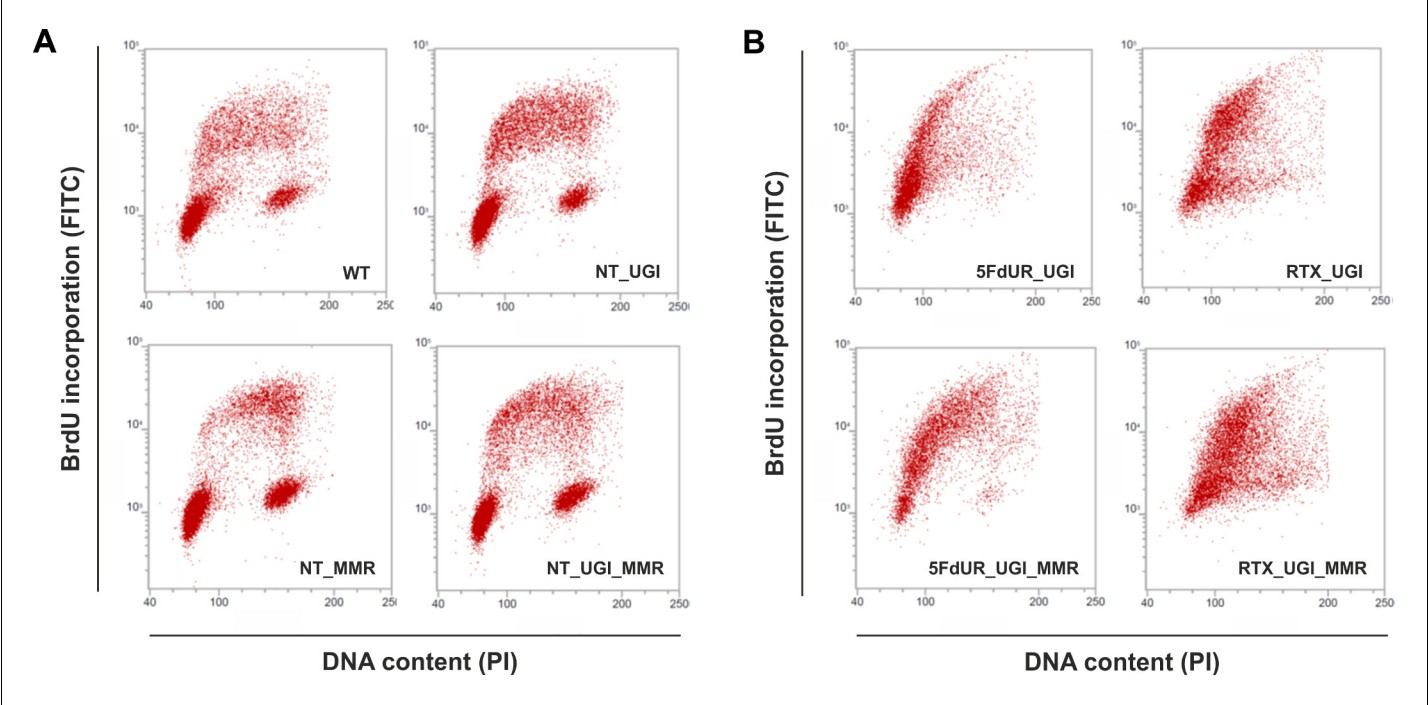

**Figure 5.** Cell cycle analysis showing the impact of UGI expression with or without drug treatments in MMR deficient and proficient HCT116 cells. Scatter plots represent the flow cytometric measurements of BrdU incorporation and propidium iodide (PI) DNA-staining. (**A**) Cell cycle distribution in non-treated, MMR deficient (WT), and UGI-expressing (NT_UGI); or in MMR proficient (NT_MMR), and UGI-expressing (NT_UGI_MMR) HCT116 cells. (**B**) Cell cycle differences caused by 5FdUR or RTX drug treatments in MMR deficient, UGI-expressing (5FdUR_UGI and RTX_UGI); or in MMR proficient, UGI-expressing (5FdUR_UGI_MMR and RTX_UGI_MMR) HCT116 cells.

The online version of this article includes the following figure supplement(s) for figure 5:

**Figure supplement 1.** Flow cytometry-based detection of γH2AX, indicative to DSBs involved in DNA damage response.

detection of genomic U-DNA in complex eukaryotic cells using microscopy. *Figure 6A* shows a schematic representation of the U-DNA staining procedure. The U-DNA sensor constructs were fused to different tags allowing antibody-based or direct detection *via* fluorescence microscopy. In order to achieve a versatile labelling technique and to facilitate super-resolution imaging of U-DNA, we attached a SNAP-tag to the C'-terminal end of ΔUNG (FLAG-ΔUNG-SNAP), generating a novel sensor construct (*Figure 6—figure supplement 1A*). The SNAP-tag offers a flexible biorthogonal chemical labelling strategy as it reacts specifically and covalently with benzylguanine derivatives, permitting the irreversible labelling of SNAP fusion proteins with a wide variety of synthetic probes (*Keppler et al., 2003*). In order to check whether the functionality of this new construct is still preserved, we performed dot blot and staining experiments. Results shown in *Figure 6—figure supplement 1B* indicate that the FLAG-ΔUNG-SNAP construct is functional and shows similarly reliable U-DNA detection using dot blot approach, when compared to FLAG-ΔUNG-DsRed protein described previously (*Róna et al., 2016*). *Figure 6—figure supplement 1C* shows that the new labelling construct, FLAG-ΔUNG-SNAP, also recognizes the presence of extrachromosomal uracil enriched plasmid aggregates in the cytoplasm. These results confirmed that the FLAG-ΔUNG-SNAP construct is capable of U-DNA detection in dot blot assays and suitable for in situ staining applications.

Our goal was to use this new sensor to detect in situ endogenous uracils in human cells in a setup that also allows colocalization with other chromatin factors. For visualization of our sensor, photostable SNAP-tag substrates (here SNAP647 or SNAP546) were used. *Figure 6B* shows that drug treatment and the inhibition of cellular UNG enzyme by UGI lead to significantly increased uracil content in genomic DNA that is readily observable on conventional confocal microscopic images. *Figure 6B* also demonstrates that our FLAG-ΔUNG-SNAP sensor can be applied for straightforward staining of genomic uracil after either Carnoy (as used previously [*Róna et al., 2016*]) or PFA fixation. Unlike

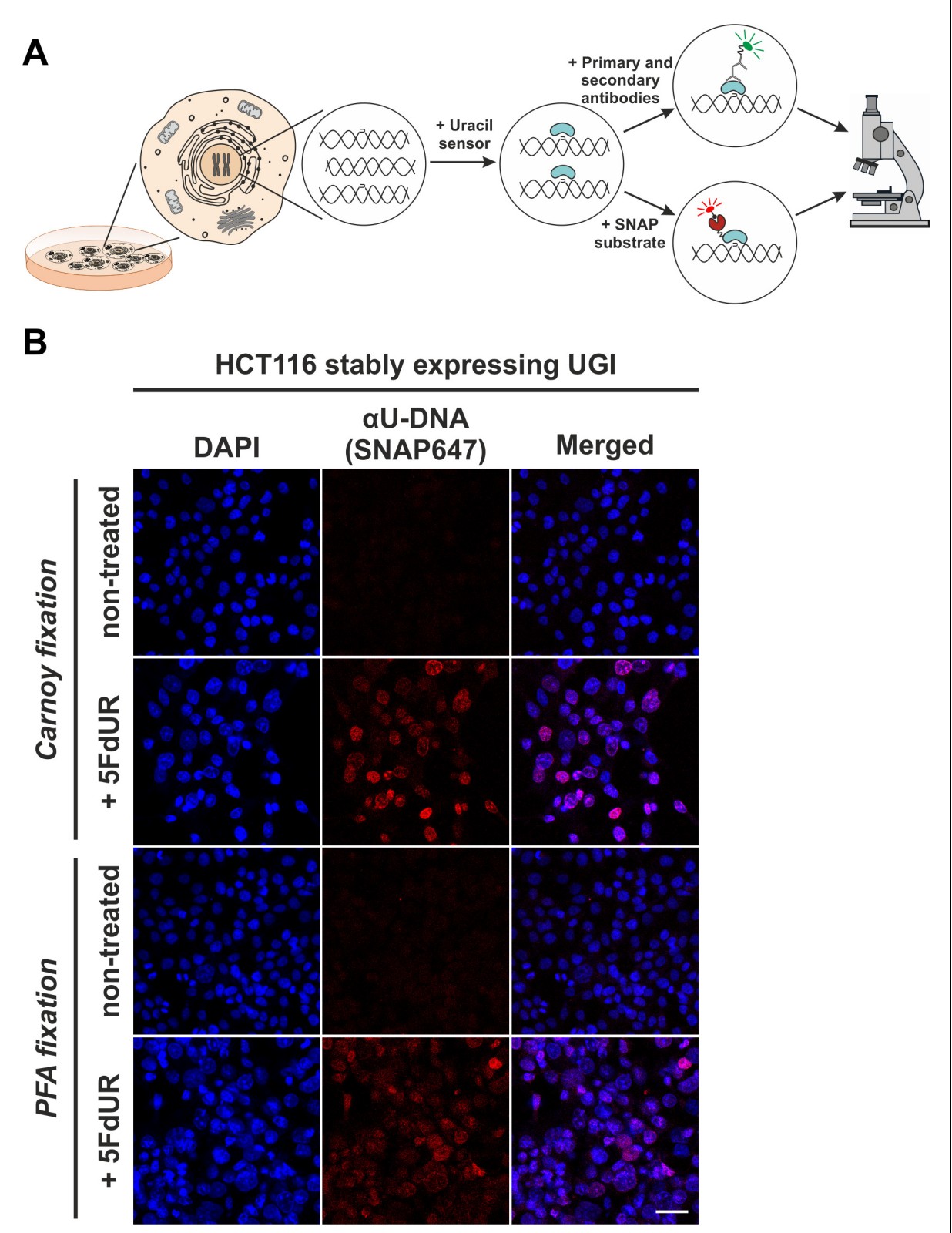

**Figure 6.** In situ detection of the cellular endogenous U-DNA content. (**A**). Scheme represents that genomic uracil residues can be visualized in situ using our further developed U-DNA sensor construct *via* immunocytochemistry (through FLAG-tag) or directly *via* SNAP-tag chemistry. (**B**) HCT116 cells expressing UGI and treated with 5FdUR show efficient staining with the uracil sensor compared to non-treated cells, detected by confocal microscopy. Uracil residues are labelled by our FLAG-ΔUNG-SNAP sensor protein visualized by the SNAP647 substrate. DAPI was used for DNA counterstaining.
*Figure 6 continued on next page*

*Figure 6 continued*

Our optimized staining method is capable of comparable, specific uracil detection in HCT116 cells even with paraformaldehyde (PFA) fixation compared to the Carnoy fixation applied previously (*Róna et al., 2016*). Scale bar represents 40 μm. Note that the nuclei of the treated cells (5FdUR_UGI) are enlarged as compared to the non-treated ones (NT_UGI) presumably due to cell cycle arrest (*Huehls et al., 2016*; *Yan et al., 2016*). The online version of this article includes the following figure supplement(s) for figure 6:

**Figure supplement 1.** Design and validation of the new FLAG-ΔUNG-SNAP uracil-sensor.

Carnoy, PFA fixative is compatible with most antibody-based staining procedures, thus it is suitable for multi-color imaging allowing colocalization studies. Next, we attempted to use super-resolution microscopy to have a better track of the uracil distribution pattern even in case of the low genomic uracil level found in the non-treated cells. *Figure 7* compares confocal, STED and dSTORM microscopy techniques for U-DNA detection. The exquisite sensitivity of dSTORM is apparent from these experiments as it can detect the low level of genomic uracil in non-treated cells (*Figure 7B*). Importantly, we observed different heterogeneous staining in the nucleus for uracil in non-treated and drug-treated cells. Furthermore, images of drug-treated cells show uracil staining with signal enrichment at the nuclear membrane and areas surrounding the nucleoli. Movies in *Figure 7—videos 1–4* (for the corresponding representative image see *Figure 7—figure supplement 1*) contribute to further visualization of uracil distribution captured by confocal and STED imaging.

Based on the genome-wide sequencing data analysis, we proceeded to select cognate chromatin markers for colocalization studies. As shown in *Figure 4A*, the highest similarity (GIGGLE) scores corresponded to H3K36me3 and H3K27me3 for the RTX and the 5FdUR treated samples, respectively. Furthermore, Segway analysis strengthened that these two histone markers (from the 22 investigated factors) show the most similar signal distribution pattern to the U-DNA patterns of drug-treated samples (*Figure 4B*). Using the herein demonstrated immunofluorescence protocol we obtained co-stained images of uracil and these histone markers by both confocal and dSTORM microscopies (*Figure 8A–B*). Validating the U-DNA-Seq data, we found that U-DNA staining shows significant colocalization with staining for both chromatin markers; H3K36me3 and H3K27me3, which was quantified using a cross-pair correlation analysis of the dSTORM images as shown in *Figure 8C–D*. The rate of colocalization, as determined by the interaction factor (IF) value (*Bermudez-Hernandez et al., 2017*; *Whelan et al., 2018*), was statistically significant between the uracil signal and both chromatin markers in each case of drug treatment, when compared to the non-treated sample as well as to a generated set of random distribution patterns of these chromatin markers. The cross-pair correlation method probes the probability distributions across all possible pair-wise distances between two species, taking in account the number of foci for each species (*Coltharp et al., 2014*; *Cutler et al., 2013*; *Veatch et al., 2012*; *Yin and Rothenberg, 2016*). This normalization of the number of foci ensures that any increase in IF is specifically due to an increase in their co-localization probability density, and not due to the increase in the amount of either species.

## Discussion

Here we focus on the alteration of U-DNA distribution pattern upon treatment with drugs perturbing thymidylate biosynthesis. Towards this end, we combined two new applications of further developed U-DNA sensor that was originally described in *Róna et al., 2016*. On one hand, using a DNA-IP-seq like application, termed U-DNA-Seq, we provided genome-wide uracil distribution data that was compared to the patterns of different genomic features. On the other hand, in immunocytochemistry, the sensor was applied to detect colocalization of U-DNA and selected histone markers.

Using U-DNA-Seq, here we demonstrate that the distribution of uracil-containing regions is altered in the drug-treated (5FdUR or RTX, in combination with UGI) as compared to the non-treated (wild type and UGI-expressing) samples. We demonstrated that UGI expression did not cause any observable change either on cell cycle progression (*Figure 5*, and *Luo et al., 2008*) or uracil distribution pattern (*Figure 3*). We chose HCT116 cancer cell line that is deficient in mismatch repair (MLH1$^{-/-}$), similarly to many types of cancer especially in colon cancer (*Germano et al., 2018*; *Gupta and Heinen, 2019*; *Sekine et al., 2017*). As its mismatch repair proficient counterpart is also

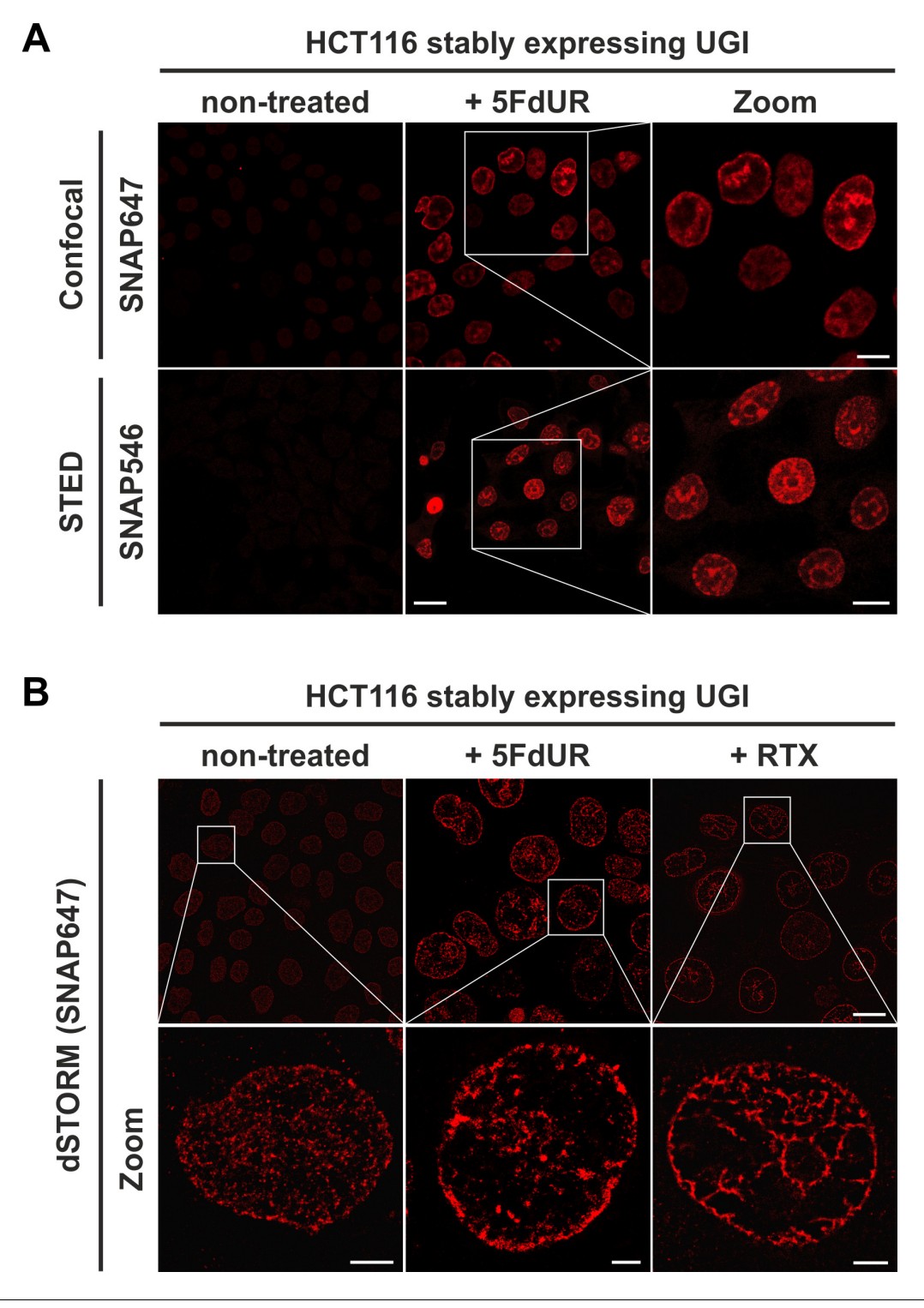

**Figure 7.** The FLAG-ΔUNG-SNAP sensor enables super-resolution detection of genomic uracil by STED and dSTORM microscopy. (**A**) U-DNA staining was performed on non-treated or 5FdUR treated HCT116 cells stably expressing UGI. Different SNAP-tag substrates, SNAP647 for confocal and SNAP546 for super-resolution imaging (STED) were used to label FLAG-ΔUNG-SNAP. Scale bar represents 20 μm for whole images and 10 μm for zoomed sections. (**B**) dSTORM imaging was performed on non-treated or drug-treated (5FdUR or RTX) HCT116 cells stably expressing UGI to compare the sensitivity of these imaging techniques. U-DNA staining shows a characteristic distribution pattern in cells with elevated uracil levels as compared to non-treated cells. SNAP647

*Figure 7 continued on next page*

*Figure 7 continued*

substrate was used to label FLAG-ΔUNG-SNAP. Scale bar represents 10 μm for whole images and 2 μm for zoomed sections.

The online version of this article includes the following video and figure supplement(s) for figure 7:

**Figure supplement 1.** Representative image for movies in *Figure 7—videos 1–4*.

**Figure 7—video 1.** Confocal Z-stack series of 3 cells showing uracil distribution.
https://elifesciences.org/articles/60498#fig7video1

**Figure 7—video 2.** 3D projection of *Figure 7—video 1*.
https://elifesciences.org/articles/60498#fig7video2

**Figure 7—video 3.** Z-stack series of 1 cell acquired by STED showing uracil distribution.
https://elifesciences.org/articles/60498#fig7video3

**Figure 7—video 4.** 3D projection of *Figure 7—video 3*.
https://elifesciences.org/articles/60498#fig7video4

---

available (*Koi et al., 1994*), we took the opportunity to address the impact of the MMR status on genomic uracil distribution. We found that the genomic uracil pattern is much more influenced by MMR proficiency in case of the 5FdUR treatment than in case of the RTX treatment (*Figure 3*). The genomic uracil distribution patterns either in non-treated or in drug-treated cells are found to be non-random: broad regions of uracil enriched genomic segments were detected. Within the third part of our pipeline (*Figure 2—figure supplement 1* and *Supplementary file 3–5*), we also analyzed the distribution pattern of these broad peaks comparing them to a set of relevant and cell type-specific data of ChIP-seq experiments and other genomic features. In drug-treated cells, these broad segments showed the highest correlation with ChIP-seq-based patterns published for predominantly euchromatin and facultative heterochromatin markers (*Figure 4A–B*). Increasing evidence suggests that active and repressed chromatin states can be determined in a combinatorial fashion where simultaneous histone marks can efficiently shift gene expression from inactive to active states or vice versa (*Gates et al., 2017*; *Hyun et al., 2017*). Hence, it is of special interest to note that our colocalization data show similarity scores not just for one but for a variety of factors. Such combinatorial behavior was further demonstrated by the genome segmentation analysis using the Segway package (*Figure 4B*) that also pointed to the fact that the distribution of histone markers are not fully matched with the detected U-DNA pattern. Hence further genomic features were also studied (*Figure 4C*). Importantly, regarding these factors and additional features, our results are highly coherent. Namely, the outstanding correlation of uracil-DNA patterns in drug-treated samples with active promoters, CpG islands, early replicating segments and DNase hypersensitive sites, all of which are published for normally cycling cells, highly supports the above conclusion. Euchromatin was shown to imply early replicating genomic regions, whereas heterochromatin replicates in late S-phase (*Black et al., 2012*). Accordingly, we report that the drug treatment induced U-DNA pattern is more similar to the early replicating segments, whereas U-DNA distribution in non-treated (wild type and UGI-expressing) cells shows simultaneous association with both heterochromatin markers and late replicating regions (*Figure 4C–D*, also supported by *Figure 4—figure supplement 3*). It has to be noted that MMR proficiency leads to a major decrease in the correlation with early replication timing in case of the 5FdUR treated sample (*Figure 4C–D*), and a smaller decrease in the correlation with transcriptionally active regions in case of both treatments (*Figure 4A–C*). Still, difference between the uracil-DNA patterns of drug-treated and non-treated samples remains unambiguous, regardless the MMR status (*Figures 3* and *4*).

Taken together, in the non-treated cells, where the level of genomic uracil is low, we show that uracil is preferentially located in the constitutive heterochromatin, which can be explained by the fact that heterochromatin is generally highly condensed and thus less accessible for DNA repair and replicative DNA synthesis. In contrast, in the open, more frequently transcribed euchromatin, DNA repair can efficiently correct uracils in the presence of a balanced dNTP pool. The low amount of genomic uracil in non-treated cells might remain from either cytosine deamination or thymine replacing misincorporation that escaped DNA repair. However, drug (5FdUR or RTX) treatments perturb the cellular nucleotide pool, and consequently highly increase the rate of thymine replacing uracil misincorporation events overwriting the background uracil pattern of non-treated cells' genome. Uracil appearance *via* thymine replacing misincorporation implies prior DNA synthesis involved in

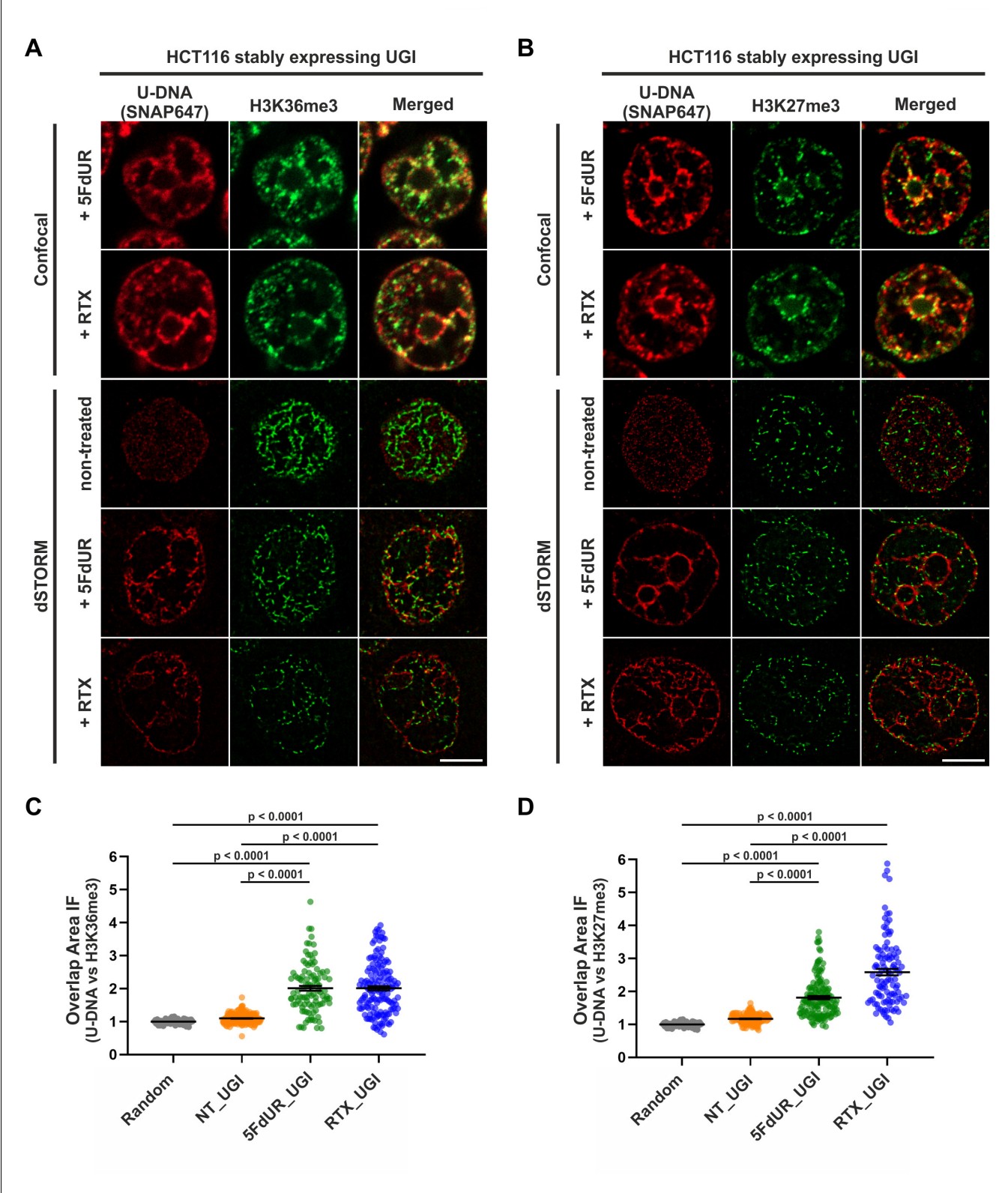

**Figure 8.** Genomic uracil moieties colocalize with H3K36me3 and H3K27me3 analyzed by super-resolution microscopy. Confocal and dSTORM imaging were performed on non-treated, 5FdUR or RTX treated HCT116 cells stably expressing UGI to compare the localization of genomic uracil residues (red) to histone markers, H3K36me3 (green) (A) or H3K27me3 (green) (B). Scale bar represents 5 μm. The graphs display the cross-pair orrelation analysis between U-DNA and H3K36me3 (C) or H3K27me3 (D) performed on dSTORM images. Overlap is defined as any amount of pixel overlap between
*Figure 8 continued on next page*

Figure 8 continued

segmented objects. Total numbers of analyzed nuclei for H3K36me3 staining (**C**) were the following: NT_UGI (n = 205), 5FdUR_UGI (n = 101) and RTX_UGI (n = 153) from two independent experiments. Total numbers of analyzed nuclei for H3K27me3 staining (**D**) were the following: NT_UGI (n = 154), 5FdUR_UGI (n = 151) and RTX_UGI (n = 107) from two independent experiments. Black line denotes the mean of each dataset, and error bars represent standard errors of the mean (SEM). The color code follows the one in *Figure 3A*. Source data are available in *Figure 8—source data 1*. The online version of this article includes the following source data for figure 8:

**Source data 1.** Interaction factors between U-DNA and selected histone marks, determined in colocalization measurements using dSTORM microscopy.

either replication, or transcription-coupled DNA repair, or epigenetic reprogramming (e. g. erasing the methyl-cytosine epigenetic mark). Importantly, we found that uracil pattern showed the highest correlation with the features (early replication, active promoters and DNase hypersensitive sites, and CpG islands) linked exactly to these processes (*Figure 4C*). This is further supported by the fact that in MMR proficient drug-treated samples higher U-DNA content was measured as compared to the MMR deficient ones (*Figure 1—figure supplement 1C–D*). This observation in MMR proficient cells might be explained by either longer segments synthesized during the MMR process (*Bowen et al., 2013*), or the less tight control on cell cycle arrest (*Figure 5*) allowing more extended replicative synthesis.

Our data showing that under normal conditions, that is in lack of drug treatment, localization of human genomic uracils can be associated with the heterochromatic regions which is in agreement with the recent study by *Shu et al., 2018*. We propose that this pattern may reflect less efficient DNA repair in the heterochromatin. In accordance, it was shown that mutation rate within the later replicating heterochromatin is markedly increased (*Stamatoyannopoulos et al., 2009*). Interestingly, uracil distribution in bacterial and yeast genomes was found to be mostly excluded from the earliest as well as from the latest replicating segments (*Bryan et al., 2014*), suggesting a partially different pattern as compared to what is observed in human cells. In yeast cells it was also shown that transcription coupled repair synthesis might result in elevated uracil incorporation into actively transcribed regions under normal conditions (*Kim and Jinks-Robertson, 2009*; *Owiti et al., 2018*). However, both yeast and bacteria show major differences in both mechanisms of dNTP pool regulation (*Mathews, 2014*) and the set of available UDGs, as they do not encode the SMUG1 enzyme which is an important backup of the UNG2 in human (*Elateri et al., 2003*; *Kavli et al., 2002*). These differences may account for the alterations found in the genomic uracil distribution patterns.The antifolate or nucleotide-based thymidylate synthase inhibitors, such as 5-FU, RTX or 5FdUR are known to lead to cell cycle arrest, as it is confirmed in our experimental system (*Figure 5*) and is also reflected in the detected uracil-DNA pattern that strongly correlates with the early replicating segments in case of both drug treatments. The two drugs caused similar, but not equivalent uracil-DNA pattern. On the one hand, the correlations with the H3K36me3 marker as well as with the early replicating segments are both markedly stronger with the RTX treated sample as compared to the 5FdUR treated sample (*Figure 4*). On the other hand, the correlation of uracil accumulation with the H3K27me3 marker and with the CpG islands is stronger in the 5FdUR treated sample. Moreover, the MMR status has markedly different influence on the resulting U-DNA pattern in case of the two drugs (*Figures 3–4*). Such differences might correspond to drug-specific mechanism of action, involving alterations in signaling processes, transcription regulation and the timing of cell cycle arrest (*Van Triest et al., 2000*). Details of these mechanisms remain obscure in the literature. Still, it is well-known that both drugs inhibit thymidylate synthase thereby facilitating dUMP incorporation into DNA, while the nucleotide analogue 5FdUR also leads to direct incorporation of 5-fluorodeoxyuridine monophosphate (FdUP) into the DNA (*Longley et al., 2003*; *Pettersen et al., 2011*). Genomic uracil and fluorouracil might have different effects on transcription and epigenetic regulation processes that could also contribute to the observed differences of the two U-DNA patterns. It should be noted that our method detects both uracil and also fluorouracil within the DNA, since the UNG enzyme binds to fluorouracil as well (*Pettersen et al., 2011*). Phenotypic differences in cell cycle progression upon the two drug treatments were also reported. The 5FdUR treatment was shown to cause an S-phase arrest in the second cycle (*Huehls et al., 2016*; *Yan et al., 2016*), while the actual time point of cell cycle arrest upon RTX treatment is still controversial (*Blackledge, 1998*; *Ding et al., 2019*; *Zhao et al., 2016*). Similarly, we also detected slightly altered cell cycle distribution patterns in case of the two drug treatments, which were differently influenced by the MMR

status (*Figure 5*). In case of the 5FdUR treatment, MMR proficiency seems to lead to a weaker S-phase arrest. This might correspond to the observed decrease in the correlation of U-DNA pattern and early replication timing (*Figure 4C*). However, equally induced DNA damage response (reported by γH2AX) was detected upon both drug treatments (*Figure 5—figure supplement 1*). Consistently with our observations, *Weeks* et al recently showed that treatment with the antifolate pemetrexed in UNG -/- human colon cancer cells led to preferential enrichment of double-strand breaks (DSBs) within highly accessible euchromatic regions, like transcription factor binding sites, origins of replication, DNase hypersensitivity regions and CpG islands (*Weeks et al., 2014*). This study did not directly address the occurrence of uracil moieties but caught the process initiated by uracil incorporation at a later stage. Still, the distribution pattern of the resulting DSBs showed similarities to our U-DNA-Seq data.

As we demonstrated here, the genome-wide uracil distribution patterns have relevance for example in case of drug-treated cancer cells. Therefore, besides the global U-DNA quantification methods (MS based [*Galashevskaya et al., 2013*], and dot blot [*Róna et al., 2016*]), NGS-based techniques also have high impact.The presented new method, termed U-DNA-Seq is a direct, feasible alternative to the recently published UPD-Seq (*Sakhtemani et al., 2019*), Excision-seq (*Bryan et al., 2014*) or dU-seq (*Shu et al., 2018*) methods, all of which rely on indirect detection requiring one or more auxiliary chemical or enzymatic step(s). Only these three methods have the potential thus far to map genome-wide distribution of uracil within isolated genomic DNA based on NGS, and only dU-seq was used in the context of human genome. One advantage of our U-DNA-Seq is that it is a direct method employing U-DNA specific binding of catalytically inactive UNG-derived sensor constructs to pull-down uracil-containing genomic DNA-fragments. In terms of resolution, only the pre-digestion Excision-seq was shown to be able to provide single-base resolution in case of smaller size genomes with high uracil content (*Bryan et al., 2014*). The resolution of other methods including our new U-DNA-Seq is limited by the fragment size of the DNA library. Importantly, single-base resolution of uracil positions has decreased relevance in most cases, considering the basically stochastic nature of uracil appearance either by incorporation as a result of drug-treatment-induced dNTP pool perturbations during DNA synthesis due to insensitivity of the polymerases, or by spontaneous cytosine deamination. Due to the stochastic processes, the actual positions of uracils are expected to be variable in every single cell. Therefore, a statistical approach has higher descriptive value about the uracil distribution in these cases. Accordingly, we constructed a novel computational pipeline (*Figure 2* and *Figure 2—figure supplement 1*) that is suitable for the description of this kind of uracil distribution patterns. We also demonstrated that the usual analysis methods designed for ChIP-seq experiment are suboptimal in this case (*Figure 3—figure supplement 2*). Moreover, re-analysis of the earlier published dU-seq data (*Appendix 1—table 1*) with the herein developed pipeline, showed very high correlation with our U-DNA-Seq data in case of comparable samples (non-treated K562 cells in both cases; and 5FdUR treated UGI expressing HCT116 vs 5FdUR treated UNG$^{-/-}$ HEK293T cells, *Appendix 1—figure 1–2*) confirming robustness and reliability of our method. However, our interpretation is markedly different regarding the preferential centromeric location of uracils that has been suggested by *Shu et al., 2018*. We analyzed the underlying reasons of this discrepancy in Appendix 1.

The new U-DNA-Seq method was shown to be reliable, robust and potent enough to gain systematic information on uracil-DNA metabolism upon drug treatments. Such information could essentially contribute to the future understanding of the mechanistic details either of cytotoxic effect induced by anti-cancer drugs, or other biological processes involving genomic uracil appearance. To this end, it is also of key importance to establish new visualization methods allowing colocalization measurements between U-DNA and other factors in highly complex eukaryotic cells.

Therefore, we further developed the U-DNA sensor to visualize genomic uracil in situ in human cells. The FLAG-ΔUNG-SNAP sensor construct and the optimized staining method presented here were successfully applied in confocal and super-resolution (STED or dSTORM) microscopies (see *Figures 6–8*). To our knowledge, there is no alternative technique published so far for in situ microscopic detection of mammalian genomic uracil. A recent paper was published reporting a similar approach, where uracil-DNA glycosylase UdgX was coupled to a fluorescent tag and applied for staining of uracils in *E. coli* DNA (*Datta et al., 2019*), however, in our previous study ΔUNG had already been proved to be potent for in situ uracil detection in the same organism (*Róna et al., 2016*). Still, the UdgX-based tool was not further extended for detection of uracils within the highly

complex chromatin of human cells. Moreover, our detection method also allows simultaneous staining for other factors in colocalization experiments, potentially providing mechanistic insight into several important biological phenomena that involve uracil-DNA. For colocalization studies, two histone markers were selected based on the U-DNA-Seq results, namely H3K36me3 and H3K27me3, which were the strongest correlating factors for RTX_UGI and 5FdUR_UGI U-DNA patterns, respectively (*Figure 4A–B*). Using dSTORM super-resolution microscopy we could confirm significant correlation of genomic uracil with both selected histone markers in drug-treated (5FdUR or RTX), UGI-expressing cells (*Figure 8*). H3K36me3 was shown to associate with actively transcribed genes (*Becker et al., 2017*; *Hyun et al., 2017*; *Pfister et al., 2014*), while H3K27me3 is the most cited marker for facultative heterochromatin (*Becker et al., 2017*; *Saksouk et al., 2015*). Strikingly, we found that H3K27me3 shows even stronger colocalization with the U-DNA pattern in case of the RTX treated sample as compared to the 5FdUR treated one, which might be indicative for RTX treatment induced chromatin remodeling at least regarding this histone modification. It is important to note that our U-DNA-Seq was compared to published data that corresponds to ChIP-seq experiments performed in non-treated cells. However, during in situ cellular colocalization studies, the drug treatment is obviously applied to both patterns (i.e. U-DNA and histone marker). With the ChIP-seq experiment for H3K36me3 performed on both RTX treated and non-treated cells, we demonstrated that such treatment-induced chromatin remodeling is not a general phenomenon, but may rather confine to certain factors (*Figure 4A–B*, and *Figure 4—figure supplement 1*). Based on these observations, we can confirm that such in silico correlation studies has a predictive potential allowing qualitative characterization, and further independent techniques are required for detailed studies. In summary, co-staining of the selected histone markers and the genomic uracil in drug-treated cells *via* dSTORM reinforced the association between uracil occurrence and transcriptionally active regions.

It has been argued that uracil accumulation may play a more decisive role in genomic instability than the induced uracil-excision repair (*Huehls et al., 2016*; *Yan et al., 2016*). Uracil in DNA may therefore be used as a key marker for estimating efficiency of chemotherapeutic drugs targeting thymidylate biosynthesis. Our presented new techniques, namely the U-DNA-Seq and the related in situ U-DNA detection methods provide key insights into the mechanism of chemotherapeutic drugs. The combination of these methods might become a highly potent approach in the future, that is to investigate the complex pattern of intra-tumor heterogeneity that is closely related to cancer progression and drug-resistance (*Stanta and Bonin, 2018*), therefore may contribute to improving clinical practice.

# Materials and methods

**Key resources table**

| Reagent type (species) or resource | Designation | Source or reference | Identifiers | Additional information |
|---|---|---|---|---|
| Gene (bacteriophage PBS2) | UGI | *Mol et al., 1995* | UniProtKB - P14739 | UNG inhibitor protein encoded in *Bacillus subtilis* bacteriophage PBS2 |
| Antibody | Anti-FLAG M2, mouse monoclonal | Sigma | F3165 | (1:10000) |
| Antibody | anti-H3K36me3, rabbit monoclonal | Cell Signaling Technologies | 4909T | (1:8000) for ICC |
| Antibody | anti-H3K27me3, rabbit monoclonal | Cell Signaling Technologies | 9733T | (1:6000) for ICC |
| Antibody | anti-γH2AX, rabbit monoclonal | Sigma | 05–636 | (1:500) for measuring DSBs in flow cytometry |
| Cell line (*Homo sapiens*) | 293T | Yvonne Jones (Cancer Research UK, Oxford, UK) | | maintained in Dulbecco's modified Eagle's medium completed with PenStrep and FBS |

*Continued on next page*

*Continued*

| Reagent type (species) or resource | Designation | Source or reference | Identifiers | Additional information |
|---|---|---|---|---|
| Cell line (*Homo sapiens*) | K562 | European Collection of Cell Cultures | | maintained in RPMI 1640 (GlutaMAX Supplement, HEPES) Medium (Gibco), completed with PenStrep and FBS |
| Cell line (*Homo sapiens*) | HCT116 | European Collection of Cell Cultures | | maintained in McCoy's 5A medium, completed with PenStrep and FBS |
| Cell line (*Homo sapiens*) | HCT116+ch3 sub-line | C. Richard Boland (Baylor University, Dallas, Texas, US) | | sub-line of HCT116: MLH1 restored, MMR is functional |
| Cell line (*Homo sapiens*) | HCT116+ hUGI/EGFP | This paper | | sub-line of HCT116: stable expressing UGI (Materials and methods: Generation of UGI-expressing stable cell lines) |
| Cell line (*Homo sapiens*) | HCT116+ch3+ hUGI/EGFP | This paper | | sub-line of HCT116+ch3: stable expressing UGI (Materials and methods: Generation of UGI-expressing stable cell lines) |
| Strain, strain background (*Escherichia coli*) | XL1-Blue | Stratagene | | |
| Strain, strain background (*Escherichia coli*) | CJ236 [*dut-, ung-*] | NEB | | *E. coli* strain for preparation of the uracil-containing DNA |
| Strain, strain background (*Escherichia coli*) | BL21(DE3) *ung-151* | Samuel E Bennett (Oregon State University, Corvallis, US) | | *E. coli* strain for expression of ΔUNG sensor constructs |
| Sequence-based reagent | actin-for | Sigma-Aldrich, *Ho et al., 2016* | | 5'-CCTCATGGCCTTGTCACAC-3' |
| Sequence-based reagent | actin-rev | Sigma-Aldrich, *Ho et al., 2016* | | 5'-GCCCTTTCTCACTGGTTCTCT-3' |
| Sequence-based reagent | pET15b-For | Sigma-Aldrich | | 5'-CATATGCTCGAGGATCCGGC-3' |
| Sequence-based reagent | pET15b-Rev | Sigma-Aldrich | | 5'-TCATCGATAAGCTTTAATGCGGT-3' |
| Sequence-based reagent | Spin-Fw | Sigma-Aldrich | | 5'- ACCGGCATAACCAAGCCTAT-3' |
| Sequence-based reagent | Spin-Rev | Sigma-Aldrich | | 5'- ACAATGCGCTCATCGTCATC-3' |
| Recombinant DNA reagent | pLGC-hUGI/EGFP | Michael D Wyatt (South Carolina College of Pharmacy, University of South Carolina, US) | | for producing sub-lines stably expressing UGI |
| Recombinant DNA reagent | pSNAPf | NEB | N9183S | to clone the FLAG-ΔUNG-SNAP construct |
| Peptide, recombinant protein | FLAG-ΔUNG-SNAP | This paper | | produced in *E. coli* BL21(DE3) ung-151 (Materials and methods: Plasmid constructs and cloning of the FLAG-ΔUNG-SNAP construct) |
| Peptide, recombinant protein | 1xFLAG-ΔUNG | *Róna et al., 2016* | | produced in *E. coli* BL21(DE3) ung-151 |
| Peptide, recombinant protein | 3xFLAG-ΔUNG | *Róna et al., 2016* | | produced in *E. coli* BL21(DE3) ung-151 |
| Peptide, recombinant protein | FLAG-ΔUNG-DsRed | *Róna et al., 2016* | | produced in *E. coli* BL21(DE3) ung-151 |
| Commercial assay, kit | Quick-DNA Miniprep Plus Kit | Zymo Research | D4069 | for genomic DNA preparation |
| Commercial assay, kit | NucleoSpin Gel and PCR Clean-up Kit | MACHEREY-NAGEL GmbH and Co. KG | 740609.25 | for IP DNA purification |
| Commercial assay, kit | NGS including library preparation | Novogene | Novaseq 6000, 20 GB, 150 PE | as a service |
| Commercial assay, kit | 5-Bromo-2'-deoxy-uridine (BrdU) Labeling and Detection Kit I | Roche, Sigma | 11296736001 | |

*Continued on next page*

*Continued*

| Reagent type (species) or resource | Designation | Source or reference | Identifiers | Additional information |
|---|---|---|---|---|
| Chemical compound, drug | Anti-FLAG M2 agarose beads | Sigma | A2220 | for U-DNA-IP |
| Chemical compound, drug | Pierce Protein A/G Magnetic Beads | Thermo Fisher Scientific | 88802 | for ChIP |
| Chemical compound, drug | 5-fluoro-2′-deoxyuridine (5FdUR) | Sigma | F0503 | Thymidylate synthase inhibitor, treatment: 20 µM, 48 hr |
| Chemical compound, drug | raltitrexed (RTX) | Sigma | R9156 | Thymidylate synthase inhibitor, treatment: 100 nM, 48 hr |
| Chemical compound, drug | SNAP-Surface Alexa Fluor 546 | NEB | S9132S | SNAP substrate for superresolution imaging |
| Chemical compound, drug | SNAP-Surface Alexa Fluor 647 | NEB | S9136S | SNAP substrate for superresolution imaging |
| Software, algorithm | ImageJ (Fiji) | National Institutes of Health | | for densitometry, and image processing |
| Software, algorithm | Huygens STED Deconvolution Wizard | Huygens Software | | superresolution image analyzing software package |
| Software, algorithm | BWA | *Li and Durbin, 2010* | | short sequencing read aligner software |
| Softwares, algorithm | deepTools package | *Ramírez et al., 2016* | | NGS data processing tools |
| Software, algorithm | bedtools package | *Quinlan and Hall, 2010* | | tools for analyzing interval files |
| Software, algorithm | GIGGLE search | *Layer et al., 2018*. | | search tool for similarity screening in large set of interval files |
| Software, algorithm | Segway software package | *Chan et al., 2018*; *Hoffman et al., 2012* | | machine learning software for genome segmentation |
| Software, algorithm | Integrated Genome Viewer (IGV) | *Thorvaldsdóttir et al., 2013*. | | tool for visualisation of many types of processed NGS data |

## Plasmid constructs and cloning of the FLAG-ΔUNG-SNAP construct

The pLGC-hUGI/EGFP plasmid was kindly provided by Michael D. Wyatt (South Carolina College of Pharmacy, University of South Carolina, US). Generation of catalytically inactive U-DNA sensor proteins (1xFLAG-ΔUNG, 3xFLAG-ΔUNG, FLAG-ΔUNG-DsRed) was described previously (*Róna et al., 2016*). pSNAPf (New England Biolabs (NEB), Ipswich, Massachusetts (MA), US) was PCR amplified with primers SNAP-Fw (5′ – TAA TGG TAC CGC GGG CCC GGG ATC CAC CGG TCG CCA CCA TGG ACA AAG ACT GCG AAA TG - 3′) and SNAP-Rev (5′ – ATA TCT CGA GGC CTG CAG GAC CCA GCC CAG G - 3′). The resulting fragments were digested by KpnI and XhoI, and ligated into the KpnI/XhoI sites of the plasmid construct FLAG-ΔUNG-DsRed (in a pET-20b vector) yielding the FLAG-ΔUNG-SNAP construct. Scheme of the used constructs is shown in *Figure 6—figure supplement 1A*. Primers used in this study were synthesized by Sigma-Aldrich (St. Louis, Missouri, US), and all constructs were verified by sequencing at Microsynth Seqlab GmbH (Göttingen, Germany). All UNG constructs were expressed in the *Escherichia coli* BL21(DE3) *ung-151* strain and purified using Ni-NTA affinity resin (Qiagen, Hilden Germany) as described previously (*Róna et al., 2016*).

## DNA isolation and purification

pEGFP-N1 plasmid (Clontech, Mountain View, California, US) was transformed into XL1-Blue [*dut+, ung+*] (Stratagene, San Diego, California (CA), US) or CJ236 [*dut-, ung-*] (NEB) *E. coli* competent cells. Cell cultures were grown for 16 hr in Luria broth (LB) media supplemented with 50 µg/ml kanamycin at 37˚C. Plasmids used in this study were purified using PureYield Plasmid Midiprep Kit (Promega, Madison, Wisconsin, US) according to the instructions of the manufacturer. XL1-Blue and CJ236 *E. coli* strains were propagated in LB media at 37˚C and were harvested at log phase. Genomic DNA of bacterial samples as well as eukaryote cells was purified using the Quick-DNA Miniprep Plus Kit (Zymo Research, Irvine, California, US) using the recommendations of the manufacturer.

## Cell culture, transient transfection and treatment of cells

The 293T cell line was a generous gift of Yvonne Jones (Cancer Research UK, Oxford, UK). The HCT116 and the K562 cell lines were purchased from the European Collection of Cell Cultures (ECACC, Salisbury, UK). The HCT116+ch3 sub-line (a kind gift from C. Richard Boland (Baylor University, Dallas, Texas, US)) is complemented with chromosome three carrying the wild type gene for hMLH1 and is competent in MMR function. 293T cells were grown in Dulbecco's modified Eagle's medium (Gibco, Life Technologies, Carlsbad, CA, US), while HCT116 and K562 cells were maintained in McCoy's 5A medium (Gibco) and RPMI 1640 (GlutaMAX Supplement, HEPES) Medium (Gibco), respectively. Media was supplemented with 50 µg/ml Penicillin-Streptomycin (Gibco) and 10% fetal bovine serum (Gibco). Cells were cultured at 37°C in a humidified incubator with 5% $CO_2$ atmosphere. All cell lines used in this study were tested for mycoplasma contamination. HCT116 cells were transfected with FuGENE HD (Promega) according to the manufacturer's recommendation. For immunocytochemistry, HCT116 cells were transfected with normal pEGFP-N1 (purified from XL1-Blue [*dut+, ung+*] *E. coli* cells) or uracil-rich pEGFP-N1 (purified from CJ236 [*dut−, ung−*] *E. coli* cells) vector as described previously (*Róna et al., 2016*). Forty hours after transfection with UGI expressing vectors, transiently transfected cells were grown for an additional 48 hr either in the absence or presence of 20 µM 5FdUR (Sigma) before collecting them for genomic DNA purification.

## Generation of UGI-expressing stable cell lines

Retroviral packaging and stable cell line generation were done as described in *Rona et al., 2018*. Briefly, 293T cells ($1.5 \times 10^6$ cells in T25 tissue culture flasks) were transfected with 1.5 µg pLGC-hUGI/EGFP, 0.5 µg pCMV-VSV-G envelope and 0.5 µg pGP packaging plasmids using Lipofectamine 3000 transfection reagent (Invitrogen, Carlsbad, CA, US) according to the manufacturer's recommendation. The supernatant, containing lentiviral particles was collected and filtered through a 0.45 µm filter (Merck Millipore, Burlington, MA, US) 36 hr after the transfection. Successfully transduced MMR deficient and proficient HCT116 cells were collected by FACS sorting for GFP-positive cells using a BD FACSAria III Cell sorter (BD Biosciences, San Jose, CA, US). UGI-expressing cells were treated with 20 µM 5FdUR or 100 nM RTX (Sigma) for 48 hr before fixation for immunocytochemistry or collecting them for genomic DNA purification described above.

## Dot blot measurements and analysis for quantification of U-DNA

Detection of the genomic uracil content by dot blot measurements were carried out using 3xFLAG-Δ UNG construct, as described earlier (*Róna et al., 2016*). Dot blot assay was used for measuring genomic uracil levels of non-treated and drug (5FdUR or RTX) treated MMR deficient and proficient HCT116 cells expressing UGI (*Figure 1—figure supplement 1B–D*), or to confirm the successful enrichment of uracil-containing DNA (*Figure 1B*), and also to compare uracil recognition specificity of the FLAG-ΔUNG-DsRed and FLAG-ΔUNG-SNAP constructs (*Figure 6—figure supplement 1B*). Densitometry was done using ImageJ (Fiji) software (National Institutes of Health, US). Analysis of the data and the calculation of the number of deoxyuridine nucleotides in the unknown genomic DNA was described before (*Molnár et al., 2018*; *Róna et al., 2016*). Briefly, the number of uracil/ million bases in the unknown samples were determined by interpolating their normalized intensities to the calibration curve of the standard. Statistical analysis of dot blot (*Figure 1—figure supplement 1C–D*) was carried out by Microsoft Excel using the non-parametric two-sided Mann-Whitney U test. Differences were considered statistically significant at p<0.005. Data presented are representative of six independent datasets (n = 6).

## DNA immunoprecipitation

After 48 hr treatment, the surface attached cells were harvested. Genomic DNA was purified by Quick-DNA Miniprep Plus Kit (Zymo Research) and eluted in nuclease-free water. 12 µg of genomic DNA was sonicated into fragments ranging between 100 and 500 base pairs (bp) (checked by agarose gel electrophoresis) with a BioRuptor (Diagenode, Liège, Belgium). 25% of the samples was saved as input, and the remaining DNA was re-suspended in the following IP buffer: 30 mM TRIS-HCl, pH = 7.4, 140 mM NaCl, 0.01% Tween-20, 1 mM ethylenediaminetetraacetic acid (EDTA), 15 mM β-mercaptoethanol, 1 mM phenylmethylsulfonyl fluoride, 5 mM benzamidine. Immunoprecipitations were carried out with 15 µg of 1xFLAG-ΔUNG construct for 2.5 hr at room temperature with

constant rotation. Anti-FLAG M2 agarose beads (Sigma) were equilibrated in IP buffer, and then added to the IP mixture for 16 hr at 4°C with constant rotation. Beads were washed three times for 10 min in IP buffer, and re-suspended in elution buffer containing 1% sodium dodecyl sulfate (SDS), 0.1 M NaHCO$_3$. Elution of uracil sensor protein binding U-DNA was done by vortexing for 5 min with an additional incubation for 20 min with constant rotation. After centrifugation (13000 rpm for 3 min), supernatant was transferred to clean tubes. This procedure was repeated with the same amount of elution buffer, and protein/DNA eluted complexes were combined in the same tube. Samples were incubated with 150 µg/ml RNAse A (Epicentre, Paris, France) for 30 min, followed by the addition of 500 µg/ml Proteinase K (Sigma) for 1 hr at 37°C for removal of RNA and proteins. Immunoprecipitated DNA was purified with NucleoSpin Gel and PCR Clean-up Kit (MACHEREY-NAGEL GmbH and Co. KG, Düren, Germany) according to the manufacturer's instructions. Densitometry analysis of agarose gel was done using ImageJ (Fiji) software for concentration calculation of fragmented DNA. Enrichment of uracil in the DNA samples was examined by dot blot assay. DNA libraries were created from the samples and then subjected to next-generation sequencing (NGS). Scheme of U-DNA-Seq is shown in *Figure 1A*.

## Controls of U-DNA-IP method

For positive control of the U-DNA-IP, uracil-containing 315 bp spike-in oligo was prepared by PCR amplification from pET15b in the presence of 0.02 mM dUTP, and 0.2 mM dNTP mix using TEMPase Hot Start DNA polymerase (VWR (Radnor, Pennsylvania, US)). Uracil-free oligo was also amplified under the same reaction conditions but in the absence of dUTP. Primer sequences are as follows: pET15b-For: 5'-CATATGCTCGAGGATCCGGC-3'; pET15b-Rev: 5'-TCATCGATAAGCTTTAATGCGG T-3'. Spike-in oligos were purified with NucleoSpin Gel and PCR Clean-up Kit. 2.5 nM uracil-containing or uracil-free spike-in DNA was added into 3 µg of sonicated genomic DNA from non-treated HCT116 cells, then DNA-IP was carried out as described above. Enrichment was measured by qPCR (on a QuantStudio 1 qPCR instrument (Thermo Fisher Scientific (Waltham, MA, US))) and calculation was based on the comparison of the Cq values for IP samples using uracil-containing and uracil-free spike-in oligos. Primer sequences are as follows: Spin-Fw: 5'- ACCGGCATAACCAAGCCTAT-3'; Spin-Rev: 5'- ACAATGCGCTCATCGTCATC-3'. For negative control of the U-DNA-IP, mock IP experiments were also performed using empty anti-FLAG beads not containing the U-DNA sensor on genomic DNA from non-treated (NT_UGI) and 5FdUR treated (5FdUR_UGI), UGI-expressing HCT116 cells, using the same protocol as described above. The amounts of pulled down DNA were much decreased in these control IPs as compared to their true IP counterparts, still NGS were performed (*Figure 1—figure supplement 2*, *Supplementary file 1*).

## High-throughput DNA sequencing and data analysis

Sequencing of input and enriched U-DNA samples were done on two independent biological replicates at BGI (China) generating 100 bp paired-end reads (PE) on a HiSeq 4000 instrument or at Novogene (China) using the Novaseq 6000 platform resulting in 150 bp PE reads. Analysis pipeline is summarized in *Figure 2*, and details including the applied command lines and scripts are found in the *Supplementary file 1* and *3–5*. Sequencing reads were aligned to the GRCh38 human reference genome (version GRCh38.d1.vd1) (*Jensen et al., 2017*) using BWA (version 0.7.17) (*Li and Durbin, 2010*). Aligned reads were converted to BAM format and sorted using samtools (version 1.9) (*Li et al., 2009*). Duplicate reads were marked using Picard Tools (version 1.95). As a part of preprocessing, blacklisting and filtering of ambiguously mapped reads were also performed (*Supplementary file 1* and *Figure 2—figure supplement 2*; *Amemiya et al., 2019*). For data processing, to derive uracil distribution signal, first, normalized coverage signals were calculated and smoothened using bamCoverage from the deepTools package (*Ramírez et al., 2016*), which resulted in genome-scaled coverage tracks in bigWig format. Then, log2 ratio of the coverage tracks (enriched/input) were calculated with bigwigCompare. These bigwig files were compared using the multiBigwigSummary, Pearson correlations were calculated using the plotCorrelation tools also from the deepTools package (*Figure 3B*). From the log2 ratio tracks, interval (bed) files were derived using reasonable thresholds (for details see *Supplementary file 1* and *Figure 3—figure supplement 2A*). Log2 ratio signal distribution (*Figure 3C*) was calculated using R. Peaks of coverage were also called using the MACS2 with broad option (version 2.1.2), a standard tool in chromatin marker ChIP-

seq data analysis (*Feng et al., 2011*; *Zhang et al., 2008*). Results of peak calling and the regions derived from the log2 ratio tracks were compared (*Figure 3—figure supplement 2*). Hereafter, the two terms 'peaks' and 'regions' will be consequently applied for the results of the two approaches, respectively. For the negative control IP samples, genome-scaled coverage tracks were also calculated in the same way. Then normalized signal tracks were subtracted from their corresponding U-DNA-IP tracks, and combined with their input to calculate log2 enrichment tracks (*Supplementary file 1* and *Figure 1—figure supplement 2*). Colocalization analysis of identified uracil enriched regions with other ChIP-seq and DNA accessibility data was performed on a dataset containing HCT116 specific or other relevant data only (for details see *Supplementary file 3*) using GIGGLE search tool (*Layer et al., 2018*). To plot results of GIGGLE search, OriginPro 8.6 was used (*Figure 4A*). Genome segmentation analysis on our U-DNA-Seq data, our H3K36me3 ChIP-seq data, and HCT116 specific ChIP-seq data from the ENCODE database was performed using Segway software package (*Chan et al., 2018*; *Hoffman et al., 2012*; *Supplementary file 3*, and *Figure 4B*). Measuring overlaps with other genomic features (*Figure 4C*) was done using bedtools annotate tool (*Quinlan and Hall, 2010*) as it is described in *Supplementary file 4*. Replication timing scores and AT content were calculated on the genomic segments defined by the Segway analysis as described in *Supplementary file 4* (*Figure 4—figure supplement 3*). Correlation analysis between uracil enrichment and replication timing (*Figure 4D* and *Appendix 1—figure 2C*) was done using R as it is described in *Supplementary file 5*. Sequencing data were visualized (*Figure 3A*, *Figure 1—figure supplement 2*, *Figure 2—figure supplement 2*, *Figure 3—figure supplement 1*, *Figure 4—figure supplement 2*, *Supplementary file 2*, *Appendix 1—figure 1A*, *Appendix 1—figure 2A*) using the IGV browser (*Thorvaldsdóttir et al., 2013*).

## Chromatin immunoprecipitation and sequencing (ChIP-seq)

Sub-confluent cultures of UGI-expressing (non-treated or treated with 100 nM RTX for 48 hr) cells were washed with phosphate-buffered saline (PBS) and cross-linked with 1% paraformaldehyde (PFA) for 10 min, then quenched with the addition of 0.15 M glycine. Cells then were rinsed with ice-cold PBS twice and lysed with buffer LB1 (50 mM TRIS, pH = 7.5, 140 mM NaCl, 2 mM EDTA, 0.5 mM EGTA, 0.5% NP-40, 0.25% Triton X-100, 10% glycerol, and protease inhibitor cocktail) for 10 min at 4°C, then in LB2 (10 mM TRIS, pH = 7.5, 200 mM NaCl, 1 mM EDTA, 0.5 mM EGTA, and protease inhibitor cocktail) for 10 min at 4°C. Nuclei pellets were sonicated in LB3 (10 mM TRIS, pH = 7.5, 0.5% N-Lauroylsarcosine sodium salt, 1 mM EDTA, 0.5 mM EGTA, and protease inhibitor cocktail) in a BioRuptor, which yielded fragments between 100 and 500 bp. After centrifugation for 10 min at 4°C, supernatants were diluted in dilution buffer (50 mM TRIS, pH = 7.5, 0.5% NP-40, 1 mM EDTA, 150 mM NaCl) followed by pre-clearing of Pierce Protein A/G Magnetic Beads (Thermo Fisher Scientific) for 3 hr at 4°C. Immunoprecipitation was performed overnight at 4°C using anti-H3K36me3 (CST (Danvers, MA, US), cat.no.: 4909T) antibody following the supplier's recommendations. After immunoprecipitation, protein A/G magnetic beads (pre-cleared with IgG-free fetal bovine serum albumin (BSA, Jackson ImmunoResearch (Cambridgeshire, UK)), overnight at 4°C) were added for further 7 hr of incubation. Precipitates were washed sequentially for 10 min each with the 1:1 combination of dilution buffer and HS buffer (20 mM TRIS, pH = 8.0, 0.1% SDS, 1% NP-40, 2 mM EDTA, 500 mM NaCl), with HS buffer, and finally with dilution buffer. Precipitates were then washed with TE buffer (10 mM TRIS-HCl, pH = 7.5, 1 mM EDTA) and eluted two times with 1% SDS and 0.1 M NaHCO$_3$. Eluates were pooled and heated overnight at 65°C to reverse the formaldehyde crosslinking. Samples were incubated with 100 µg/ml RNAse A for 30 min, then with 200 µg/ml Proteinase K for 1 hr at 37°C for removal of RNA and proteins. Immunoprecipitated DNA was purified with NucleoSpin Gel and PCR Clean-up Kit according to the manufacturer's instructions. Quantitative PCR analysis for human β-actin was carried out to check the efficiency of the H3K36me3 IP using the following primer sequences: actin-for: 5'-CCTCATGGCCTTGTCACAC-3'; actin-rev: 5'-GCCCTTTCTCACTGGTTCTCT-3' (*Ho et al., 2016*). DNA libraries were created from the samples and then subjected to NGS at Novogene using the Novaseq 6000 platform resulting in 150 bp PE reads. Data analysis were performed similarly to the U-DNA-Seq analysis (*Figure 4—figure supplement 1*), details are provided in *Supplementary file 3*.

## Cell cycle analysis and γH2AX staining

2D cell cycle analysis was performed using 5-Bromo-2′-deoxy-uridine (BrdU) Labeling and Detection Kit I (Roche, Sigma) and Propidium Iodide (PI, Sigma) staining (*Figure 5*). Non-treated or drug-treated (20 µM 5FdUR or 100 nM RTX for 48 hr) HCT116 cells were labelled with 10 µM BrdU for 20 min followed by trypsinization, PBS washing and overnight fixation in 70% ethanol at 4°C. DNA was denatured for 30 min with 2 M HCl, 0.5% Triton X-100. Cells were re-suspended in 0.1 M sodium tetraborate (pH = 8.5) for 10 min, and then washed with blocking buffer (1% BSA, 0.05% Tween-20 in PBS). Samples were incubated with anti-BrdU antibody (1:10) in blocking buffer for 30 min at room temperature. After washing, Ig fluorescein coupled (FITC) anti-mouse (1:10) secondary antibody was applied in blocking buffer for 30 min. Finally, after a washing step, cells were incubated with propidium iodide (10 µg/ml) and RNase A (20 µg/ml) for 30 min in PBS. Occurrence of DSBs was investigated by immunofluorescent staining of γH2AX (*Figure 5—figure supplement 1*). Briefly, non-treated or drug-treated cells were fixed in 70% ethanol (overnight at 4°C), then DNA was denatured for 30 min with 2 M HCl, 0.5% Triton X-100. After blocking, cells were stained with an antibody against γH2AX (1:500, Sigma, cat.no.: 05–636) overnight at 4°C. FITC anti-mouse secondary antibody (1:10) was added for 30 min. Cell cycle analysis and measurement of γH2AX levels were carried out by flow cytometry with a BD FACSCalibur Cell Analyzer.

## Immunofluorescent staining of uracil residues

Detection of uracil residues was done in extrachromosomal plasmids after transfection (*Figure 6—figure supplement 1C*) or in genomic DNA of HCT116 cells (*Figures 6–8*). Staining of extrachromosomal DNA was done as described previously (*Róna et al., 2016*) with minor modifications for comparison of FLAG-ΔUNG-DsRed or FLAG-ΔUNG-SNAP sensor constructs. Briefly, uracil residues were visualized by applying 1.5 µg/ml of the FLAG-ΔUNG-DsRed or the FLAG-ΔUNG-SNAP, and then primary (anti-FLAG M2 antibody (1:10000, Sigma)) and secondary antibodies (Alexa 488 (1:1000, Molecular Probes, Eugene, Oregon, US)). For immunofluorescent staining of genomic uracil residues, control or HCT116 cells stably expressing UGI were seeded onto 24-well plates containing cover glasses or onto µ-Slides (or their glass bottom derivative) (ibidi GmbH, Germany) suitable for use in STED and single molecule applications, and treated as indicated. In case of dSTORM imaging, coverslips were coated with poly-D-lysine (Merck Millipore) before seeding the cells. Sub-confluent cultures of cells were fixed using 4% PFA (pH = 7.4 in PBS) or Carnoy's fixative (ethanol: acetic acid: chloroform = 6:3:1) for 15 min. In case of dSTORM imaging, cells were pre-extracted with ice-cold CSK buffer (10 mM PIPES, pH = 6.8, 100 mM NaCl, 300 mM sucrose, 1 mM EGTA, 3 mM MgCl$_2$, 0.25% Triton X-100) containing protease and phosphatase inhibitor tablets (Roche, Basel, Switzerland) for 5 min before PFA fixation. After washing or rehydration steps (1:1 ethanol:PBS, 3:7 ethanol: PBS, PBS), epitope unmasking was done by applying 2 M HCl, 0.5% Triton X-100 for 30 min. DNA denaturation with HCl was required in order to increase DNA accessibility for efficient staining and to eliminate any potential interaction between the overexpressed UGI and the applied UNG sensor construct. After neutralization with 0.1 M Na$_2$B$_4$O$_7$ (pH = 8.5) for 5 min followed by PBS washes, cells were incubated in blocking solution I (TBS-T (50 mM TRIS-HCl, pH = 7.4, 2.7 mM KCl, 137 mM NaCl, 0.05% Triton X-100) containing 5% non-fat dried milk) for 15 min, followed by incubation in blocking buffer I supplemented with 200 µg/ml salmon sperm DNA (Invitrogen) for an additional 45 min. Uracil residues were visualized by applying 4 µg/ml of the FLAG-ΔUNG-SNAP construct for 1 hr in blocking buffer I with 200 µg/ml salmon sperm DNA at room temperature. After several washing steps with TBS-T containing 200 µg/ml salmon sperm DNA, primary, then secondary antibodies were operated in blocking buffer II (5% fetal goat serum (FGS), 3% BSA and 0.05% Triton X-100 in PBS). Anti-FLAG M2 antibody (1:10000, Sigma), then Alexa 488 conjugated secondary antibody (1:1000, Molecular Probes) was applied for 1 hr in blocking buffer II, enabling visualization of FLAG epitope. SNAP-tag substrates were also used to label SNAP-tag fusion proteins when FLAG-ΔUNG-SNAP was applied as the uracil sensor protein. Cells were labelled with 2.5 µM (0.5 µM for dSTORM imaging) SNAP-Surface Alexa Fluor 546 or 647 (indicated as SNAP546 and SNAP647 in this study) (NEB) for 20 min, and optionally counterstained with 1 µg/ml DAPI (4′,6-diamidino-2-phenylindole, Sigma) nucleic acid stain, followed by several PBS washing steps before embedding in FluorSave Reagent (Calbiochem, Merck Millipore). For labelling of histone markers, anti-H3K36me3 (1:8000, CST, cat.no.: 4909T) or anti-H3K27me3 (1:6000, CST, cat.no.: 9733T) primary antibodies were used,

then visualized by Alexa 568 conjugated secondary antibody (1:10000, Molecular Probes) in dSTORM or Alexa 555 conjugated secondary antibody (1:2000, Molecular Probes) in confocal imaging.

## Confocal and STED imaging and analysis

Confocal images were acquired on a Zeiss LSCM 710 microscope using a 20x (NA = 0.8) or a 63x (NA = 1.4) Plan Apo objective or a Leica TCS SP8 STED 3X microscope using a 100x (NA = 1.4) Plan Apo objective. STED images were acquired on the Leica TCS SP8 STED 3X microscope using 660 nm STED (1.5 W, continuous wave) laser for depletion (in combination with Alexa 546). The same image acquisition settings were applied on each sample for comparison. A moderate degree of deconvolution was applied to the recorded STED images using the Huygens STED Deconvolution Wizard (Huygens Software), based on theoretical point spread function (PSF) values. Fluorescence images processed using ZEN and ImageJ (Fiji) software. 3D projection movies (*Figure 7—videos 1–4*) were constructed from Z-stack images captured by confocal or STED imaging.

## dSTORM imaging and image reconstruction

Super-resolution images were obtained and reconstructed as previously described (*Rona et al., 2018*). Briefly, dSTORM images were recorded using an in-house built imaging platform based around an inverted microscope. Two color imaging was carried out sequentially on samples labelled with SNAP-Surface Alexa Fluor 647 and Alexa Fluor 568. The imaging buffer, consisting of 1 mg/ml glucose oxidase, 0.02 mg/ml catalase, 10% glucose, 100 mM mercaptoethylamine (MEA) in PBS, was mixed and added just before imaging. For display purposes, super-resolution images shown in the manuscript have been adjusted for brightness and smoothed; however, quantitative analysis were performed on images before being manually processed to avoid any user bias.

## Interaction factor

The interaction factor (IF) quantifies the colocalization of red and green foci within a cell nucleus by measuring the area of overlap between the two sets of foci (*Bermudez-Hernandez et al., 2017*; *Whelan et al., 2018*). The positions of the green foci are then randomized and the overlap between the two colors is measured again. This randomization is repeated 20 times, and the interaction factor is the ratio between the experimental overlap area and the mean of the randomized overlap areas. If the red and green foci were completely independent of each other, the IF value would equal one. A value greater than one signifies a higher degree of colocalization compared to a random sample. Non-parametric Mann-Whitney U test was used to calculate statistics on the graphs. Differences of the IF values were considered statistically significant at $p < 0.0001$ as indicated in *Figure 8C–D*. Data are presented from two independent biological experiments.

## Acknowledgements

We gratefully acknowledged the kind help of György Török and László Homolya in acquiring fluorescent images *via* STED microscopy. We also wish to say sincere thanks to György Várady in FACS sorting and flow cytometry experiments, to Gábor Tusnády for providing access to computational capacity, and to Balázs Ligeti for providing storage and computational capacity by the infrastructure of PPKE ITK. We acknowledge the ENCODE Consortium (*ENCODE Project Consortium, 2012*) and the ENCODE production laboratory(s) generating the particular dataset(s) as well as the contributors of the UCSC Table Browser (*Karolchik et al., 2004*; *Kuhn et al., 2013*) data. We also acknowledge the contributors of Ensembl (*Zerbino et al., 2018*), ReplicationDomain (*Weddington et al., 2008*), Cistrome Data Browser (*Mei et al., 2017*) for making their data publicly available.

## Additional information

### Funding

| Funder | Grant reference number | Author |
|---|---|---|
| National Research, Development and Innovation Office of | K119493 | Beáta G Vertessy |

| | | |
|---|---|---|
| Hungary | | |
| National Research, Development and Innovation Office of Hungary | NVKP_16-1-2016-0020 | Beáta G Vertessy |
| National Research, Development and Innovation Office of Hungary | 2017-1.3.1-VKE-2017-00002 | Beáta G Vértessy |
| National Research, Development and Innovation Office of Hungary | 2017-1.3.1-VKE-2017-00013 | Beáta G Vértessy |
| National Research, Development and Innovation Office of Hungary | VEKOP-2.3.2-16-2017-00013 | Beáta G Vértessy |
| National Research, Development and Innovation Office of Hungary | NKP-2018-1.2.1-NKP-2018-00005 | Beáta G Vértessy |
| National Research, Development and Innovation Office of Hungary | NVKP_16-1-2016-0037 | Balázs Győrffy |
| National Research, Development and Innovation Office of Hungary | 2018-1.3.1-VKE-2018-00032 | Balázs Győrffy |
| National Research, Development and Innovation Office of Hungary | KH-129581 | Balázs Győrffy |
| Ministry of Human Capacities | BME FIKP-BIO | Beáta G Vertessy |
| Cancer Research UK | C37/A18784 | Carolina Gemma |
| National Institutes of Health | R35-GM136250 | Michele Pagano |

The funders had no role in study design, data collection and interpretation, or the decision to submit the work for publication.

### Author contributions

Hajnalka L Pálinkás, Conceptualization, Formal analysis, Investigation, Visualization, Methodology, Writing - original draft, Project administration, Writing - review and editing; Angéla Békési, Conceptualization, Data curation, Software, Formal analysis, Validation, Investigation, Visualization, Writing - original draft, Writing - review and editing; Gergely Róna, Conceptualization, Investigation, Visualization, Writing - original draft, Writing - review and editing; Lőrinc Pongor, Data curation, Software, Formal analysis; Gábor Papp, Data curation, Software, Formal analysis, Visualization; Gergely Tihanyi, Michael J Morten, Investigation; Eszter Holub, Formal analysis; Ádám Póti, Formal analysis, Visualization; Carolina Gemma, Simak Ali, Resources; Eli Rothenberg, Michele Pagano, Supervision; Dávid Szűts, Balázs Győrffy, Supervision, Writing - review and editing; Beáta G Vértessy, Conceptualization, Resources, Supervision, Funding acquisition, Writing - original draft, Project administration, Writing - review and editing

### Author ORCIDs

Angéla Békési ![ORCID] https://orcid.org/0000-0003-2294-3002
Gergely Tihanyi ![ORCID] http://orcid.org/0000-0003-3729-6709
Carolina Gemma ![ORCID] http://orcid.org/0000-0002-4890-9972
Simak Ali ![ORCID] http://orcid.org/0000-0002-1320-0816
Eli Rothenberg ![ORCID] http://orcid.org/0000-0002-1382-1380
Michele Pagano ![ORCID] http://orcid.org/0000-0003-3210-2442
Beáta G Vértessy ![ORCID] https://orcid.org/0000-0002-1288-2982

### Decision letter and Author response

Decision letter https://doi.org/10.7554/eLife.60498.sa1

Author response https://doi.org/10.7554/eLife.60498.sa2

# Additional files

## Supplementary files

• Supplementary file 1. Detailed analysis pipeline – methods of U-DNA-Seq data analysis. List of the investigated samples (table 1); list of applied tools (table 2); pre-processing including blacklisting and additional statistics (table 3); and methods to determine uracil enrichment pattern. All applied processing steps are given in generalized command lines.

• Supplementary file 2. IGV views of log2 ratio and regions of uracil enrichment on all the chromosomes.

• Supplementary file 3. Genome-wide analysis of uracil-DNA pattern comparing to ChIP-seq data and DNA accessibility data using either GIGGLE search or the Segway genome segmentation tool. Database information, applied command lines, full results of GIGGLE search (table 1), details of our own ChIP-seq (table 2), and list of files for Segway analysis (table 3) are provided.

• Supplementary file 4. Genome-wide analysis of uracil-DNA pattern comparing to other genomic features using bedtools annotate. Database information, applied command lines, full results (table 1), and calculation of replication timing scores and AT content on genomic segments (from the Segway analysis) are provided.

• Supplementary file 5. Detailed comparison of U-DNA pattern to replication timing data (R script).

• Transparent reporting form

## Data availability

Sequencing data have been deposited into the Gene Expression Omnibus (GEO) under accession number GSE126822 and GSE153407, which have been unified under SuperSeries GSE153408. In the following Genome Browser session, we included all the log2 coverage ratio (bigwig) and the derived uracil enriched interval (bed) files corresponding to this manuscript. The color code and the names are the same as used here. https://genome.ucsc.edu/s/bekesiangi/GSE126822_UCSC_Genome_Browser_session. Source data have been provided for Figure 1-figure supplement 1, Figure 2-figure supplement 2, Figure 3, Figure 3-figure supplement 4, Figure 4, Figure 4-figure supplement 3, Figure 8, Appendix 1-figure 1, Appendix 1-figure 2.

The following datasets were generated:

| Author(s) | Year | Dataset title | Dataset URL | Database and Identifier |
|---|---|---|---|---|
| Pálinkás HL, Békési A, Pongor L, Holub E, Papp G, Gemma C, Ali S, Győrffy B, Vértessy BG | 2020 | Genome-wide alterations of uracil distribution patterns in human DNA upon chemotherapeutic treatments | https://www.ncbi.nlm.nih.gov/geo/query/acc.cgi?acc=GSE126822 | NCBI Gene Expression Omnibus, GSE126822 |
| Pálinkás HL, Békési A, Vértessy BG | 2020 | H3K36me3 ChIP-seq in non-treated and raltitrexed treated UGI-expressing HCT116 cells | https://www.ncbi.nlm.nih.gov/geo/query/acc.cgi?acc=GSE153407 | NCBI Gene Expression Omnibus, GSE153407 |

The following previously published dataset was used:

| Author(s) | Year | Dataset title | Dataset URL | Database and Identifier |
|---|---|---|---|---|
| Shu X, Lu Z, Yi C | 2018 | Genome-wide mapping reveals that deoxyuridine is enriched in the human centromeric DNA | https://www.ncbi.nlm.nih.gov/geo/query/acc.cgi?acc=GSE99011 | NCBI Gene Expression Omnibus, GSE99011 |

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

# Appendix 1

## Re-analysis and reinterpretation of dU-seq data published in *Shu et al., 2018*

Shu and co-workers recently published the dU-seq method and reported centromeric location of genomic uracil in non-treated cells (*Shu et al., 2018*). In this study, dU-seq as a short-read sequencing technology was used to map centromeric localization of a genomic feature. Centromeres are known to be highly repetitive, poorly mappable regions of the human genome, even if centromeric model sequences (*Miga et al., 2014*) are implemented into the GRCh38 (hg38) assembly (*Guo et al., 2017*). Therefore, blacklisting and mappability filters are highly recommended to be used, especially if the analysis is focusing on such critical regions, like in this case.

The dU-seq data analysis pipeline, as it was published in the Shu et al, included the following preprocessing steps: (1) pre-alignment of the 150 bp paired-end sequencing data to the spike-in sequences applied in their experiments; (2) trimming the adaptors and the low quality segments; (3) alignment of the remaining reads to reference human genome GRCh38 (it is not clear, to which set exactly) using bowtie2. There was no mention either about (1) deduplication of the data, or (2) filtering for uniquely mapped reads, or (3) applying recommended blacklists. However, in widely accepted ChIP-seq pipelines, only uniquely mapped reads are considered as valid information (*Qin et al., 2016*).

The detection of uracil enrichment within these aligned reads was done by peak calling using MACS2, separately for the uracil 'pull-down' and the 'control' samples. It was not mentioned if the corresponding input was included as a control in the peak calling process, or not. It is also not clear what options and parameters were used in their MACS2 runs (e. g. broad, no-model or broad-cutoff). Then they subtracted the peaks detected in the 'control' from the ones called in the 'pulled-down' samples. We claim that this approach is clearly suboptimal considering the lower descriptive value and the lower reproducibility of peak calling for the uracil-DNA distribution as shown in *Figure 3—figure supplements 1–2*. To judge the reproducibility of peak calling in dU-seq data, is also not trivial because they did not uploaded all the peak data for their replicates (GSE99011). We calculated the Jaccard indices for their uploaded non-treated HEK293 (two replicates) and UNG-knockout HEK293 (four replicates) data, and found really low values (0.063 for the two replicates, and 0.030, 0.059, 0.075, 0.029, 0.029, 0.055 in the pairwise comparison of the four replicates). Jaccard indices between the individual replicates and the united data of the non-treated HEK293 sample were 0.134 and 0.092. These values are even lower than in case of similar peak calling on our U-DNA-Seq data (*Figure 3—figure supplement 2B*).

The correlation with the CENPA bound genomic regions, published as a key point in their paper, is also questionable. Raw reads of CENPA ChIP-seq data (GSE45497) were downloaded and mapped to GRCh38. Details are not provided in the paper; however, they most probably used the same procedure as in case of the dU-seq data, namely, aligning reads without blacklisting and mapping quality filters, followed by peak calling. Moreover, the CENPA data were originally aligned to GRCh37 (hg19) reference genome using a more careful algorithm filtering out potential artefacts (*Hayden et al., 2013*). It would have been recommended to either follow their approach or simply lift over the original alignment from GRCh37 to GRCh38.

Unfortunately, Shu et al incompletely uploaded the peak calling results for the replicates to the GEO database. The uploaded peaks were not always correlating with the ones they published on the manuscript's figures (*Figure 2a*, where they show a K562 peak on chromosome 21 that is actually not present in their uploaded peak data, and another peak on chromosome 16 that corresponds to aligned reads characterized with MAPQ = 0, at least in our alignment). Given missing data uploads, it is challenging to reproduce their results.

Still, we were curious, whether their dU-seq data itself (not the interpretation of that) correlates with our U-DNA-Seq data, or not. Therefore, we used our novel analysis pipeline described in the present study, to process and re-analyses raw data from *Shu et al., 2018*, notably the following data:

> K562 „input": SRR5572773/GSM2630035 and SRR5572774/GSM2630036
> K562 „control": SRR5572775/GSM2630037 and SRR5572776/GSM2630038
> K562 „PD": SRR5572777/GSM2630039 and SRR5572778/GSM2630040
> HEK293T UNGKO 5FU „input": SRR5998406/GSM2769605 and SRR5998407/GSM2769606
> HEK293T UNGKO 5FU „control": SRR5998408/GSM2769607 and SRR5998409/GSM2769608

HEK293T UNGKO 5FU „PD": SRR5998410/GSM2769609 and SRR6026694/GSM2769610

We remapped the fastq reads to the human reference genome GRCh38 (GRCh38.d1.vd1.fa, GDC reference, https://gdc.cancer.gov/about-data/data-harmonization-and-generation/gdc-reference-files) using BWA, filtered out ambiguously mapped reads, and applied a blacklist created as in case of our data (*Figure 2—figure supplement 2*). The statistics of this pre-processing are summarized in *Appendix 1—table 1*. The relative amount of the spike-in sequences is significantly higher in mock pull-down („control") and the non-treated K562 pull-down samples, where the sample DNA itself was not labelled or was present at extremely low level, respectively. The overall depth of sequencing was lower in these cases, which decrease the reliability or reproducibility of further data analysis.

**Appendix 1—table 1.** Statistics on pre-processing of dU-seq data.

Samples from the study of *Shu et al., 2018* (non-treated K562 (K562), and 5FdUR treated UNG-knock-out HEK293 cells (5FdUR_UNGKO HEK293)) are compared to our samples (5FdUR treated UGI-expressing HCT116 cells (5FdUR_UGI), and non-treated wild type K562 cells (K562), the same data as in *Supplementary file 1*-table 3). In case of dU-seq samples, inputs are genomic DNA fragmented and treated according to the dU-seq protocol, also containing additional spike-in sequences; controls are pulled-down in a mock experiment excluding UNG treatment; while PD means the pull-down samples according to the dU-seq protocol. Number of raw reads means read number before starting alignment (the sum of the mapped and unmapped reads). Uniquely mapped read means that MAPQ is not zero. The dU-seq and the U-DNA-Seq samples markedly differ in the ratio of mapped (*) and unmapped (*) reads due to the spike-in DNA applied in dU-seq only.

| Sample | Replicates | Number of raw reads | Number of mapped* reads | Unmapped* reads | | Uniquely mapped reads | | Uniquely mapped reads after blacklisting | |
|---|---|---|---|---|---|---|---|---|---|
| | | | | Number | % | Number | % | Number | % |
| K562 input | SRR5572773 | 9,59,22,009 | 9,06,63,272 | 52,58,737 | 5.48 | 8,40,45,278 | 87.62 | 8,21,92,353 | 85.69 |
| | SRR5572774 | 9,50,30,662 | 8,98,16,930 | 52,13,732 | 5.49 | 8,33,06,810 | 87.66 | 8,14,57,577 | 85.72 |
| K562 Control | SRR5572775 | 7,63,94,870 | 4,61,89,867 | 3,02,05,003 | 39.54 | 4,13,85,042 | 54.17 | 4,04,05,241 | 52.89 |
| | SRR5572776 | 7,89,62,287 | 4,10,53,994 | 3,79,08,293 | 48.01 | 3,62,60,497 | 45.92 | 3,53,93,512 | 44.82 |
| K562 PD | SRR5572777 | 8,74,66,276 | 5,41,13,837 | 3,33,52,439 | 38.13 | 4,84,46,026 | 55.39 | 4,73,24,075 | 54.11 |
| | SRR5572778 | 8,24,99,155 | 5,29,29,849 | 2,95,69,306 | 35.84 | 4,75,55,693 | 57.64 | 4,64,66,915 | 56.32 |
| 5FdUR_UNGKO | SRR5998406 | 12,56,31,380 | 10,83,20,783 | 1,73,10,597 | 13.78 | 10,10,27,428 | 80.42 | 9,90,12,730 | 78.81 |
| HEK293 input | SRR5998407 | 7,03,49,101 | 6,10,39,638 | 93,09,463 | 13.23 | 5,69,70,384 | 80.98 | 5,58,07,073 | 79.33 |
| 5FdUR_UNGKO | SRR5998408 | 11,36,54,134 | 6,26,79,292 | 5,09,74,842 | 44.85 | 5,53,33,969 | 48.69 | 5,41,29,569 | 47.63 |
| HEK293 Control | SRR5998409 | 12,91,96,940 | 5,80,03,222 | 7,11,93,718 | 55.10 | 4,97,06,846 | 38.47 | 4,86,00,418 | 37.62 |
| 5FdUR_UNGKO | SRR5998410 | 8,00,35,762 | 6,79,39,558 | 1,20,96,204 | 15.11 | 6,34,53,866 | 79.28 | 6,21,84,497 | 77.70 |
| HEK293 PD | SRR6026694 | 6,62,42,483 | 5,63,03,837 | 99,38,646 | 15.00 | 5,26,53,804 | 79.49 | 5,15,98,007 | 77.89 |
| 5FdUR_UGI input | 5FdUR1_son | 12,87,06,895 | 12,86,69,770 | 37,125 | 0.03 | 12,24,76,766 | 95.16 | 11,85,58,597 | 92.12 |
| | 5FdUR1_son | 20,19,26,203 | 20,15,60,665 | 3,65,538 | 0.18 | 19,30,86,643 | 95.62 | 18,47,56,297 | 91.50 |
| 5FdUR_UGI enriched | 5FdUR1_IP | 15,05,96,242 | 15,05,22,522 | 73,720 | 0.05 | 14,45,54,269 | 95.99 | 14,15,82,874 | 94.01 |
| | 5FdUR2_IP | 13,86,51,760 | 13,84,10,833 | 2,40,927 | 0.17 | 13,32,00,761 | 96.07 | 12,85,84,894 | 92.74 |
| K562 input | K562_son | 10,61,37,622 | 10,58,75,437 | 2,62,185 | 0.25 | 10,03,26,105 | 94.52 | 9,75,04,876 | 91.87 |
| K562 enriched | K562_IP | 10,94,90,393 | 10,93,06,854 | 1,83,539 | 0.17 | 10,53,10,296 | 96.18 | 10,21,17,055 | 93.27 |

dU-seq and U-DNA-Seq were performed in completely independent laboratories, even on different continents; applying different conditions; in case of drug-treated samples different cell lines; and obviously different experimental protocols. Still, the resulting log2 ratio tracks are in surprisingly good correlation, if we use our robust analysis pipeline. This is demonstrated in *Appendix 1—figure 1* showing an IGV view, the Pearson correlation analysis, and the histograms of uracil enrichment signal distribution, following the scheme of *Figure 3* for better comparison. The clear difference between drug-

treated and non-treated samples is also obvious. Furthermore, we have demonstrated that the centromeric peaks published in *Shu et al., 2018* localize in blacklisted area (*Appendix 1—figure 2*). However, the re-analyzed dU-seq data could confirm our interpretation on genomic uracil distribution in both non-treated and drug-treated cells, using the herein developed robust analysis pipeline.

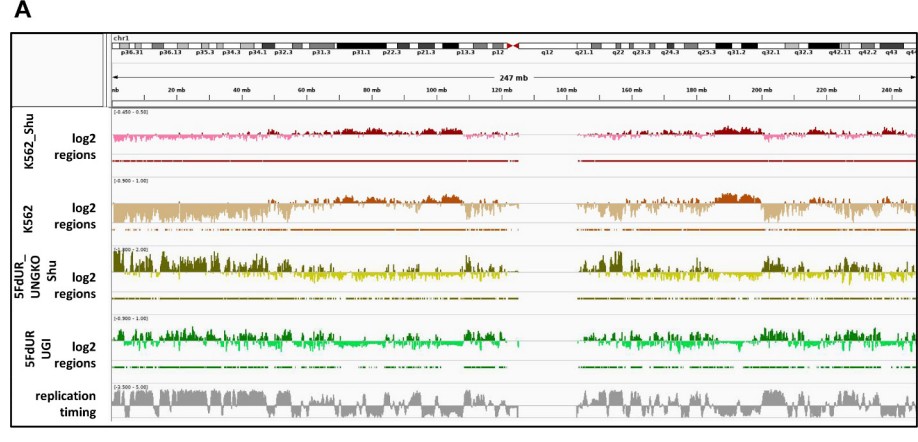

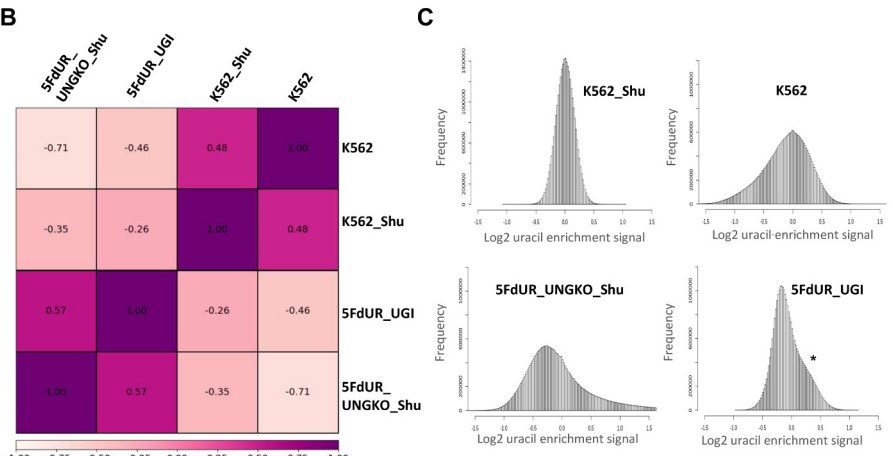

**Appendix 1—figure 1.** Re-analysis of the published dU-seq data (*Shu et al., 2018*) reveals that corresponding samples from dU-seq and U-DNA-Seq show similar patterns of uracil distribution. (**A**) IGV view of dU-seq data (non-treated K562 cells (K562_Shu, wine track), and 5FdUR treated UNG-knock-out HEK293 cells (5FdUR_UNGKO_Shu, olive track)), compared to our own U-DNA-Seq data (non-treated K562 cells (K562, brown track), and 5FdUR treated UNG inhibited HCT116 cells (5FdUR_UGI, green track)) on chromosome 1. Log2 ratio tracks and the derived regions of uracil enrichment are also indicated. The bottom track shows replication timing data (grey) for HCT116 downloaded from Replication Domain database (*Weddington et al., 2008*). (**B**) Pearson correlation among dU-seq and U-DNA-Seq log2 ratio tracks calculated from merged replicates. The drug-treated and non-treated samples are well separated again. Pearson correlation between corresponding dU-seq and U-DNA-Seq samples are unexpectedly high, especially considering the cell line difference in case of drug-treated cells, and the overall low signal intensity in case of non-treated K562. (**C**) Log2 ratio signal distribution of dU-seq and U-DNA-Seq data. The non-treated K562 samples result in a normal like distribution of uracil enrichment signals, while in case of 5FdUR treated cells, these distributions show asymmetry: either a clear shoulder (asterisk), or a more elongated tail towards increased signals in both U-DNA-Seq and dU-seq data, respectively. Source data are available in *Appendix 1—figure 1—source data 1*.

The online version of this article includes the following source data is available for figure 1:

**Appendix 1—figure 1—source data 1.** Comparison of histograms for the U-DNA signal distributions between dU-seq and U-DNA-Seq data.

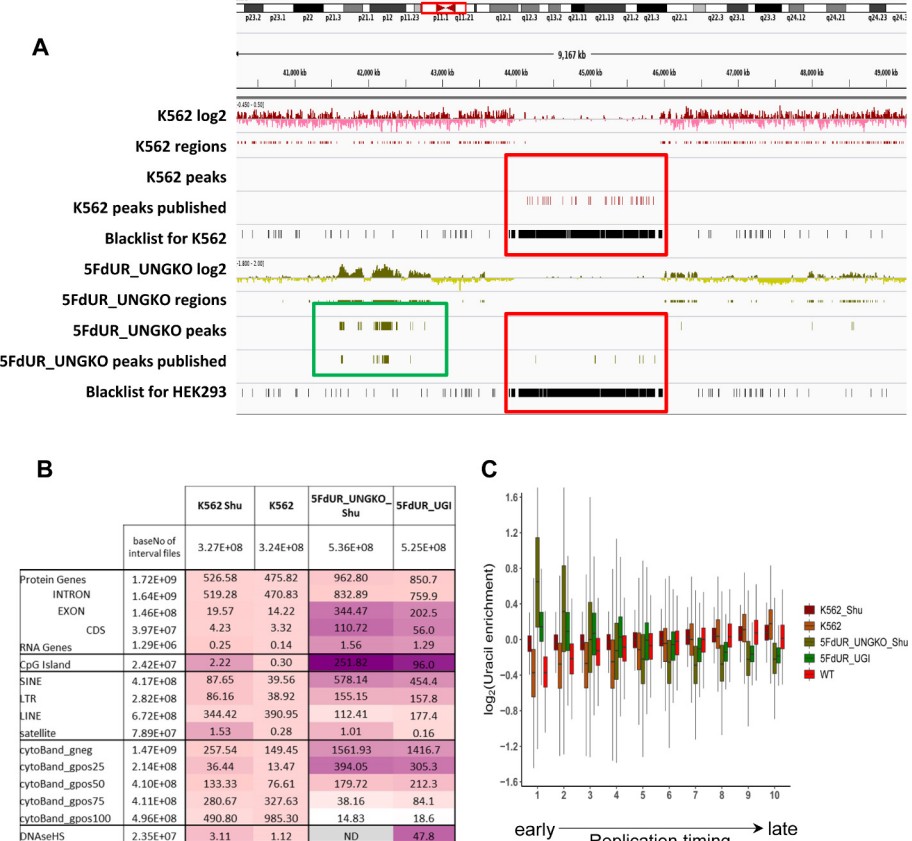

**Appendix 1—figure 2.** Reinterpretation of dU-seq data. (**A**) Comparison of the re-analyzed and the published dU-seq data in a representative IGV view of chromosome 8 zoomed to the centromeric region (also Figure 3 in *Shu et al., 2018*). All centromeric peaks in K562 published for chromosome 8 in Shu et al are found in blacklisted regions (red box). Overall, 75% of their published peaks in K562 are overlapping with blacklisted regions determined by our protocol (*Figure 2—figure supplement 2*). Accordingly, no peaks were called in the presented region during the re-analysis of the sequencing data (red box). Similarly, in drug-treated samples, published centromeric peaks were not reproducible (red box), while other peaks outside of the centromeres were similar in the published and the re-analyzed data (green box). (**B**) dU-seq data shows similar correlation to genomic features as compared to the corresponding U-DNA-Seq data. Similarity were measured by bedtools annotate tool and the scores were calculated in the same way as it was in *Figure 4C*. For each sample, cell type (HCT116 or K562) specific DNase hypersensitive site data were used. For 5FdUR treated HEK293 cells, similarity to DNase hypersensitive site data was not addressed (grey). (**C**) Correlation between uracil distribution and replication timing were confirmed by dU-seq data as well, although this correlation is weaker than the U-DNA-Seq results (*Figure 4D*). Replication timing data (bigWig files with 5000 bp binsize) were downloaded from ReplicationDomain database: Int90617792 for HCT116; Int57383924 for HEK293; Int37482971 for K562. Source data are available in *Appendix 1—figure 2—source data 1*.

The online version of this article includes the following source data is available for figure 2:

**Appendix 1—figure 2—source data 1.** Comparison of dU-seq and U-DNA-Seq data regarding correlation between U-DNA patterns and replication timing.

