## [Decision Letter]

**Acceptance summary:**

Your manuscript describes the application of combination of novel approaches – one molecular and the other microscopic, to map the location of uracil in the human genome in cells treated with the drugs 5-FdUR and RTX. The significant finding showing the correlation between a specific set of epigenetic markers and the distribution of uracil should be of great interest to the field of DNA repair and genome instability. Additionally, these findings provide significant leads for the mechanism of how the nucleotide pool imbalance caused by chemotherapeutics such 5-FdUR or RTX leads to genotoxic effects.

**Decision letter after peer review:**

[Editors’ note: the authors submitted for reconsideration following the decision after peer review. What follows is the decision letter after the first round of review.]

Thank you for choosing to send your work, "Genome-wide alterations of uracil distribution patterns in human DNA upon chemotherapeutic treatments", for consideration at *eLife*. Your article has been reviewed by three peer reviewers, one of whom is a member of our Board of Reviewing Editors, and the evaluation has been overseen by Jessica Tyler as the Senior Editor. The reviewers have opted to remain anonymous.

Our decision has been reached after consultation between the reviewers. Based on these discussions and the individual reviews below, we regret to inform you that your work will not be considered at this moment for publication in *eLife*. However, we believe that this is a very interesting study that after further work would be appropriate for *eLife*. It reveals that uracil distribution in DNA after the treatment with inhibitors of thymidine synthesis pathway is non-random. The pattern of enrichment of uracils at actively transcribed regions is very intriguing and anticipates many interesting questions regarding the mechanism. The correlation between uracil distribution and the histone markers associated with active chromatin is also very interesting, but the epigenetic marks have to be determined again under the conditions of this experimental system. The technique used to confirm such correlation, high-resolution microscopy, is very novel and promising. However, as detailed in the reviews, many of the controls that would add rigor to such techniques and findings are lacking. With the sufficient controls to support the main findings of the paper, this work would be highly significant reporting of novel findings and approaches. Please note that our policy is to aim to publish articles with a single round of revision that would typically be accomplished within two months. In our judgment, this manuscript would need extensive additional work beyond such time limit. So we would invite to resubmit a properly revised version attending the comments of the 3 reviewers accompanying your submission with a rebuttal explaining point-by-point the comments of the reviewers.

Reviewer #1:

In this manuscript, a catalytically inactive form of uracil DNA glycosylase is used to probe the genome-wide distribution patter of uracil in human cells. The expression of mutant UNG construct to detect uracil in DNA has been previously reported by the same group in Rona et al., 2016. In the same previous paper, IF was carried out to show that this construct is capable of detecting uracil-containing DNA in *E. coli* cells or on plasmids in MEF. This manuscript goes further from the previous report by analyzing the uracil distribution pattern in human cells following treatment with drugs (5FdUR and RTX) that inhibit thymidylate synthase enzyme and thus disrupt the dUTP-dTTP balance. Several different variations of the mutant UNG construct are used in two distinct major approaches in this manuscript. The first is to carry out ChIP-seq with antibody against the mutant UNG protein that will bind to uracil in DNA. The second is to modify the construct for the super-resolution immunofluorescent experiments to visualize the regions of uracil in DNA. The first approach was used to show that U-DNA are enriched at transcribed regions and early replicating regions, among others. The pattern of U-DNA in the genome after treatment with 5FdUR or RTX was also compared to the distribution pattern of multiple transcription factors and histone modification markers. Similar conclusion was made that histone markers unique to actively transcribed regions showed most similar pattern of distribution compared to U-DNA after drug treatment. The latter approach with IF was used to confirm that uracil in genomic DNA co-localize with two histone markers H3K36me3 and H3K27me3. Also, the potential mechanism of such distribution pattern is discussed. These are important findings that can also add to the understanding of the mechanism of the cytotoxicity of TS-inhibiting drugs. Overall, since the very interesting technical approach largely overlaps with the previously published report, the significance of the current manuscript should be in the substantive support for the new and significant finding regarding the distribution pattern of uracil after the drug treatment. There are however a few major flaws that detracts from the solidness of the significance of those findings. Two major weak points that call for further experiments are detailed below.

1) Figure 4. For the GIGGLE analysis, the ChIP-seq data for the histone markers and transcription factors were assembled from previously published data. The data set were matched for the same cell type used by the authors for U-DNA sequencing (HCT116). However, there was no consideration given to the fact that the treatment with 5FdUR or RTX could alter the pattern of transcription factors/histone markers significantly, obscuring the GIGGLE analysis between these data sets. Although generating a complete set of ChIP-seq data for all the different factors/markers for the 5FdUR or RTX-treated HCT116 cells would be beyond the scope of this manuscript and is not expected, a confirmation of the analysis for a number of the top hits would make for a much stronger argument. Generation of ChIP-seq results for H3K36me3 in RTX treated cells would complement the IF co-localization data shown in Figure 7 and more importantly would add to convince that such IF co-localization could reach the resolution to state confidently that U-DNA and the region of H3K36me3-modification overlap.

2) Figure 7: For the co-localization assay, H3K36me3 and H3K27me3 were chosen for further study. The rationale given by the authors is "As shown in Figure 4A, the highest similarity (GIGGLE) score corresponded to H3K36me3 and H3K27me3 for the RTX and 5FdUR treated samples." But according to Figure 4A, the correlation appears to be clearer for the H3K27ac. Why H3K27ac was not chosen instead for further study by IF should be discussed. Most notably, the negative correlation with WT or untreated sample was not very distinctive for the H3K27me3, making the difference in GIGGlE scores between NT_UGI and RTX_UGI very small. This is somewhat conflicting with the significant difference noted in the "Overlap area IF" between NT_UGI and RTX_UGI in Figure 7D and requires explanation. In general, αU-DNA signal for the 5FdUR or RTX-treated samples are much stronger than non-treated samples and whether difference in the signal strength interferes with the calling of the co-localization (overlap) is not fully discussed or controlled for. For convincing argument that super-resolution imaging can be used to analyze the genomic distribution pattern of uracil, similar experiments with at least one negative marker (such as H3K9me3 in Figure 4A) should be carried out.

Reviewer #2:

This manuscript describes a promising new low-resolution method for the visualization and mapping of uracils in DNA. It uses UNG inhibitor UGI in combination with a TYMS inhibitor Raltitrexed (RTX) or 5FdUR to increase uracils in the genome and then uses a mutant UNG to pull-down and sequence the DNA fragments. The authors claim that their data suggest that the increased uracils are predominantly in euchromatin/early replicating regions based on epigenetic markers. While the method has several attractive features including an ability to visualize uracil foci (especially Figure 7), there are several serious shortcomings of the manuscript.

1) The use of HCT116 cell line and UGI are troubling. The lack of mismatch repair (MMR) in HCT116 may affect their results. It is possible that the lack of active MMR during early replication affects the removal of uracils and thymines misincorporated across guanines. While UGI binds UNG and inhibits its action, it is unknown whether it has other physiological effects in the cells. In the very least, the results of this study should be compared with those in which MMR+ derivatives of HCT116 cells are used and UNG KO cells lines are compared with their WT parents.

2) Their uracil mapping software lacks a proper negative control. They calculate log2 ratio and plot this ratio across the genome for treated and untreated cells. The input DNA has not undergone the pull-down and hence the DNA pulled down using anti-FLAG antibodies will always show a different genomic distribution than the input DNA. The U-DNA sequencing results from treated cells should be normalized with respect to U-DNA sequencing reads from the pull-down of untreated samples, not the input DNA.

3) This method is unable to pinpoint uracils in the DNA that is pulled down. This is because, the uracils are in U:A pairs and hence they cannot be distinguished from normal T:A pairs after pull-down and sequencing. This is in contrast with most of the other available methods that can map uracils to a specific base pair. For this reason, this is a low resolution alternative to existing uracil mapping methods.

4) The manuscript lacks a proper positive control. It may be useful for the authors to introduce uracils by other means such introduction of AID gene to introduce uracils in the Ig locus and then demonstrate that their method can detect such localization of uracils. In the absence of a positive control, it is hard to know whether their method works correctly. It is possible- for example- that the treatment of cells with RTX or 5FdUR arrests the cells in early S phase and that is why the uracil incorporation occurs in early replicating regions. In which case, the lack of uracil-enriched fragments from the late replicating heterochromatin regions is a somewhat trivial result.

5) The authors dismiss the dU-seq (Shu et al., 2018) rather lightly. Shu and colleagues verified their findings by enrichment of centromeric regions showed by LC-MS/MS that centromeric uracils increased in UNG-knockout cells. Finally, they reintroduced UNG in the cells and showed that this reversed the uracil localization to centromeres. As noted above, the authors of the current manuscript have not done such careful controls in their experiments.

In summary, this is a moderately useful new technique to map uracils that does not distinguish itself from the other existing techniques except in the area of direct visualization of uracils in cells. If the authors were to use this unique feature of their method to answer an existing question, it would be attractive.

Reviewer #3:

The authors develop a new approach to measure the location and density of incorporated uracils in the human genome. They identify an interesting pattern of uracil incorporation in the human genome under normal and thymidine-depleted (using TS inhibitors) conditions, and associate these patterns with specific chromatin features. They also generalize the tool to visualize uracil incorporation with fluorescence microscopy. Both of these are significant advances of use to the DNA repair community. Moreover, these data may eventually be used to probe molecular and cellular consequences caused by common chemotherapies.

1) This study would be improved by employing computational methods designed to identify patterns across multiple independent genomic signals. The one-by-one correlations the authors present provide some insight into general correlations between uracil density, replication timing, and histone modifications, but this ad hoc approach could miss important correlations across all these signals. The segway software (Hoffman, Noble, et al) was developed specifically for this purpose and could be applied to data they have in hand to understand broader genome-wide correlations.

2) More characterization of any DNA damage response in these cells would be helpful to understand whether human uracil incorporation patterns are a "neutral" pattern in the absence of DDR (e.g., similar to high levels of seemingly innocuous uracil incorporation found in dut ung *E. coli* and yeast) , or represent a stress-induced condition caused by uracil incorporation followed by some checkpoint activation. Some flow cytometry would be useful to see the effect of RTX on cycling of these cells, and H2AX phosphorylation could be measured in bulk by western and in the microscopy experiments upon treatment.

3) Given previous indications that replication timing is somehow linked to uracil incorporation, more discussion is warranted to reconcile similarities and differences in this relationship in *E. coli*, yeast, and human uracil incorporation patterns relative to replication timing and associated changes in nucleotide pools.

[Editors’ note: further revisions were suggested prior to acceptance, as described below.]

Thank you for submitting your article "Genome-wide alterations of uracil distribution patterns in human DNA upon chemotherapeutic treatments" for consideration by *eLife*. Your article has been reviewed by three peer reviewers, one of whom is a member of our Board of Reviewing Editors, and the evaluation has been overseen by Jessica Tyler as the Senior Editor. The reviewers have opted to remain anonymous.

The reviewers have discussed the reviews with one another and the Reviewing Editor has drafted this decision to help you prepare a revised submission.

Summary:

In this paper, the authors probe the genome-wide distribution patter of uracil in human cells using two innovative approaches. First, a catalytically inactive form of uracil DNA glycosylase is expressed and subjected to ChIP-seq to generate the genomic map of uracil distribution. The mutant UDG is also used in a super-resolution immuno-fluorescence experiments for higher throughput confirmation of uracil distribution pattern. These two complimentary approaches represent significant and useful advances in understanding the genome instability induced by uracil in DNA. Here, the authors' specific interest is the uracil distribution pattern in human cells following treatment with drugs (5FdUR and RTX) that inhibit thymidylate synthase enzyme and thus disrupt the dUTP-dTTP balance. The major findings are that U-DNA are mostly enriched at transcribed regions and early replicating regions. The correlation between the U-DNA in 5FdUR- or RTX-treated cells with epigenetic markers unique to actively transcribed regions, first determined using the ChIP-seq data, was further confirmed through the super resolution IF experiments. Such correlation is significant because it can inform the potential mechanism of uracil incorporation into the genome in thymine-depleted cells and thereby add to the understanding of the mechanism of the cytotoxicity of TS-inhibiting drugs.

Revisions:

The following minor revisions are strongly recommended for acceptance to publish in *eLife*.

1) The criticism of other methods are given the disproportionate significance in the discussion. Although the effort to highlight the advance and innovation of the technique used in the current manuscript over other previously published methods is completely valid and helpful, overly long and highly critical discussion of other methods diminishes the meaningful discussion of the findings in this paper and becomes unnecessary distraction. There is also a risk of misstating what the other methods do or point out deficiencies that may have simple explanations. A separate review paper comparing various methods would better suit such discussion. I recommend that the authors reorganize the Discussion section so that it starts by describing what this manuscript has accomplished and briefly summarize the pros and cons of all the methods towards the end.

2) Clarification is required regarding the replication scores derived from Weddington et al. Is the distribution of scores correlated with timing (negative = early, positive = late?). It's unfortunate that ENCODE Repli-seq data is not available for the HCT116 cell line; this would have been a better comparison. Please add discussion or speculate on the possible mechanism regarding why uracil enrichment upon treatment both appear at similar replication timing, relative to the control samples (Figure 4—figure supplement 3)?

---

## [Author Response]

[Editors’ note: the authors resubmitted a revised version of the paper for consideration. What follows is the authors’ response to the first round of review.]

It reveals that uracil distribution in DNA after the treatment with inhibitors of thymidine synthesis pathway is non-random. The pattern of enrichment of uracils at actively transcribed regions is very intriguing and anticipates many interesting questions regarding the mechanism. The correlation between uracil distribution and the histone markers associated with active chromatin is also very interesting, but the epigenetic marks have to be determined again under the conditions of this experimental system.

We performed a pair of ChIP-seq experiments for H3K36me3 epigenetic mark (as reviewer #1 suggested) in non-treated and RTX-treated UGI-expressing HCT116 cells. We found that there are no major differences in the genome-wide distribution of this factor, or in its correlation with the U-DNA-Seq patterns upon the applied treatment conditions. Results are presented in Figure 4—figure supplement 1, and Figure 4A-B. Corresponding texts are inserted to the respective figure legends (4A: “GIGGLE scores are also indicated for our own H3K36me3 ChIP-seq…”, 4B: “…our own ChIP-seq data for H3K36me3…”); to the Results (a paragraph started with “In order to decide whether drug treatments may cause any notable changes…”); to the Materials and methods (a paragraph with the title: “Chromatin immunoprecipitation and sequencing (ChIP-seq)”), to the Discussion (“With the ChIP-seq experiment for H3K36me3…”); and to the Supplementary file 3 (a section with the title: “ChIP-seq for H3K36me3 histone marker in non-treated and RTX treated UGI-expressing HCT116 cells.” including Supplementary file 3—table 2). ChIP-seq peak results are also uploaded to the GEO GSE126822.

The technique used to confirm such correlation, high-resolution microscopy, is very novel and promising. However, as detailed in the reviews, many of the controls that would add rigor to such techniques and findings are lacking. With the sufficient controls to support the main findings of the paper, this work would be highly significant reporting of novel findings and approaches.

In the revised manuscript, we provide a positive control experiment for the U-DNA-Seq, namely, a uracil-containing spike-in sequence was added to genomic DNA fragments, then the U-DNA-IP was performed and the enrichment of the spike-in was followed by qPCR. Corresponding sentences are inserted to the Results (“To further confirm the capability of U-DNA-IP, uracil-containing spike-in DNA was combined with…”), and to the Materials and methods (a paragraph with the title: “Controls of U-DNA-IP method”).

In the submitted manuscript, we have already mentioned a negative control experiment for the U-DNA-Seq: “Specificity of U-DNA immunoprecipitation is also underlined by the fact that pull-down with empty anti-FLAG beads not containing the U-DNA sensor resulted in negligible amount of DNA (less than 5%).” During the revision, we have obtained the full genome sequencing data for this negative (empty bead) control, and we provide the corresponding data in Figure 1—figure supplement 2, and Supplementary file 1—table 1. The main text was also modified accordingly, see text segment: “Still, genome-wide sequencing data could be obtained from these negative control samples as well. We demonstrated that subtracting such control signals (for details see Supplementary file 1) will not affect the detected uracil distribution pattern regardless if the sample was drug-treated or not (Figure 1—figure supplement 2B-C). These control experiments provided confidence about the applicability and specificity of our U-DNA-IP method.”. Corresponding description was added to the Materials and methods (a paragraph with the title: “Controls of U-DNA-IP method” and “High-throughput DNA sequencing and data analysis”).

Reviewer #1:In this manuscript, a catalytically inactive form of uracil DNA glycosylase is used to probe the genome-wide distribution patter of uracil in human cells. The expression of mutant UNG construct to detect uracil in DNA has been previously reported by the same group in Rona et al., 2016. In the same previous paper, IF was carried out to show that this construct is capable of detecting uracil-containing DNA in *E. coli* cells or on plasmids in MEF. This manuscript goes further from the previous report by analyzing the uracil distribution pattern in human cells following treatment with drugs (5FdUR and RTX) that inhibit thymidylate synthase enzyme and thus disrupt the dUTP-dTTP balance. Several different variations of the mutant UNG construct are used in two distinct major approaches in this manuscript. The first is to carry out ChIP-seq with antibody against the mutant UNG protein that will bind to uracil in DNA. The second is to modify the construct for the super-resolution immunofluorescent experiments to visualize the regions of uracil in DNA. The first approach was used to show that U-DNA are enriched at transcribed regions and early replicating regions, among others. The pattern of U-DNA in the genome after treatment with 5FdUR or RTX was also compared to the distribution pattern of multiple transcription factors and histone modification markers. Similar conclusion was made that histone markers unique to actively transcribed regions showed most similar pattern of distribution compared to U-DNA after drug treatment. The latter approach with IF was used to confirm that uracil in genomic DNA co-localize with two histone markers H3K36me3 and H3K27me3. Also, the potential mechanism of such distribution pattern is discussed. These are important findings that can also add to the understanding of the mechanism of the cytotoxicity of TS-inhibiting drugs. Overall, since the very interesting technical approach largely overlaps with the previously published report, the significance of the current manuscript should be in the substantive support for the new and significant finding regarding the distribution pattern of uracil after the drug treatment. There are however a few major flaws that detracts from the solidness of the significance of those findings. Two major weak points that call for further experiments are detailed below.1) Figure 4. For the GIGGLE analysis, the ChIP-seq data for the histone markers and transcription factors were assembled from previously published data. The data set were matched for the same cell type used by the authors for U-DNA sequencing (HCT116). However, there was no consideration given to the fact that the treatment with 5FdUR or RTX could alter the pattern of transcription factors/histone markers significantly, obscuring the GIGGLE analysis between these data sets. Although generating a complete set of ChIP-seq data for all the different factors/markers for the 5FdUR or RTX-treated HCT116 cells would be beyond the scope of this manuscript and is not expected, a confirmation of the analysis for a number of the top hits would make for a much stronger argument. Generation of ChIP-seq results for H3K36me3 in RTX treated cells would complement the IF co-localization data shown in Figure 7 and more importantly would add to convince that such IF co-localization could reach the resolution to state confidently that U-DNA and the region of H3K36me3-modification overlap.

We agree to the possibility that treatment with 5FdUR or RTX could alter the pattern of transcription factors/histone markers. In our manuscript, we use the comparison between the uracil-DNA distribution pattern in normal and drug-treated cells and the pattern of the epigenetic markers reported in the databases for the same cell line (HCT116) under normal conditions as a potential indication that need to be investigated further by independent experiments. We use super-resolution microscopy (dSTORM) as a relevant and highly potent further independent technique. We found that dSTORM qualitatively confirmed the positive correlation between the epigenetic markers and the U-DNA patterns (reported in the GIGGLE scores). We also agree that determination of the epigenetic marker distribution under drug treatment would be very interesting and informative and this is one of the directions we plan to continue our further research.

We agree with the reviewer’s comment that “generating a complete set of ChIP-seq data for all the different factors/markers for the 5FdUR or RTX-treated HCT116 cells would be beyond the scope of this manuscript”. Instead, the reviewer suggested to generate ChIP-seq data for H3K36me3 in RTX treated cells, as an example to “complement the IF co-localization data”. We agree to this suggestion and in our revision, we have performed the required ChIP-seq experiments for non-treated UGI-expressing and RTX-treated UGI-expressing cells. We found that there are no major differences in the genome-wide distribution of this factor, or in its correlation with the U-DNA-Seq patterns upon the applied treatment conditions. Results are presented in Figure 4—figure supplement 1, and Figure 4A-B. Corresponding texts are inserted to the respective figure legends (4A: “GIGGLE scores are also indicated for our own H3K36me3 ChIP-seq…”, 4B: “…our own ChIP-seq data for H3K36me3…”); to the Results (a paragraph started with “In order to decide whether drug treatments may cause any notable changes…”); to the Materials and methods (a paragraph with the title: “Chromatin immunoprecipitation and sequencing (ChIP-seq)”), to the Discussion (“With the ChIP-seq experiment for H3K36me3…”); and to the Supplementary file 3 (a section with the title: “ChIP-seq for H3K36me3 histone marker in non-treated and RTX treated UGI-expressing HCT116 cells.” including Supplementary file 3—table 2). ChIP-seq peak results are also uploaded to the GEO GSE126822.

2) Figure 7: For the co-localization assay, H3K36me3 and H3K27me3 were chosen for further study. The rationale given by the authors is "As shown in Figure 4A, the highest similarity (GIGGLE) score corresponded to H3K36me3 and H3K27me3 for the RTX and 5FdUR treated samples." But according to Figure 4A, the correlation appears to be clearer for the H3K27ac. Why H3K27ac was not chosen instead for further study by IF should be discussed. Most notably, the negative correlation with WT or untreated sample was not very distinctive for the H3K27me3, making the difference in GIGGlE scores between NT_UGI and RTX_UGI very small. This is somewhat conflicting with the significant difference noted in the "Overlap area IF" between NT_UGI and RTX_UGI in Figure 7D and requires explanation. In general, αU-DNA signal for the 5FdUR or RTX-treated samples are much stronger than non-treated samples and whether difference in the signal strength interferes with the calling of the co-localization (overlap) is not fully discussed or controlled for. For convincing argument that super-resolution imaging can be used to analyze the genomic distribution pattern of uracil, similar experiments with at least one negative marker (such as H3K9me3 in Figure 4A) should be carried out.

The main reason for our selections of the H3K36me3 and the H3K27me3 markers for IF colocalization measurements was that these factors produced the highest scores for RTX and 5FdUR treatments, respectively, as reviewer #1 also wrote. We agree that we could have selected other factors like H3K27ac, H3K9ac, H3K4me3, or even the transcription factor SP1, where both treatments resulted in high GIGGLE scores and these are well separated from the non-treated cases. However, with the choice of H3K36me3 and the H3K27me3, we sought to cover better the spectra from the actively transcribing euchromatin to the facultative heterochromatin. Furthermore, with this choice, we could also address putative differences between the two drug treatments. A new genome segmentation analysis performed in our revision upon the request of the reviewer #3, also supports our choice (cf. Figure 4B in the revised manuscript, and inserted text to the Results (“Furthermore, Segway analysis strengthened…”)).

Regarding the reviewer’s comments that “the negative correlation with WT or untreated sample was not very distinctive for the H3K27me3…”, and that “this is somewhat conflicting with the significant difference noted in the "Overlap area IF" between NT_UGI and RTX_UGI in Figure 7D…”, we provide the following explanation. On the one hand, as the reviewer also pointed out (point #1), the two experimental setup has a clear difference, namely in GIGGLE analysis we compare our uracil patterns in non-treated and drug-treated cells with non-treated ChIP-seq data from databases, while in dSTORM microscopy the overlap of the two features can be calculated in the same treated cells. On the other hand, the available ChIP-seq data are from different experiments and experimental backgrounds, hence show quite big deviation among themselves. Therefore, such in silico correlation analysis can provide potential indications, but not strict rules for the choice of markers to be addressed in colocalization studies by dSTORM microscopy. Still, the in silico analysis gave meaningful prediction, as co-staining of the selected histone markers and the genomic uracil in drug-treated cells reinforced the association between uracil occurrence and transcriptionally active regions. Moreover, with the H3K36me3 ChIP-seq (cf. point #1), we also demonstrated that the treatment induced chromatin remodeling is not a general phenomenon, but may rather confines to certain histone marks. The apparent conflict between our results from these different techniques is further explained now in the Discussion of the revised manuscript (“Strikingly, we found that H3K27me3 shows even stronger colocalization with the U-DNA pattern in case of the RTX treated sample…”).

Regarding the possible influence of the overall signal strength on the calculation of IF (overlap), we have inserted a detailed explanation to the Results: “The cross-pair correlation method probes the probability distributions across all possible pair-wise distances between two species, taking in account the number of foci for each species (PMIDs: 25179006, 23717596, 22384026 and 27545293). This normalization of the number of foci ensures that any increase in IF is specifically due to an increase in their co-localization probability density, and not due to the increase in the amount of either species.”

Regarding the suggestion to stain for another marker (such as H3K9me3) that is supposed to negatively correlate with genomic uracils based on the GIGGLE scores and also on the genome segmentation results (cf. Figure 4A-B), we agree that it would be interesting to perform such an experiment. However, both Figure 4 and Figure 4—figure supplement 3 show that the distribution pattern of the genomic uracils is not fully represented by any of the histone markers. The best correlating features are the early replication timing and AT rich heterochromatin in case of the drug-treated and non-treated samples, respectively. As a negative control for the drug-treated samples, it would be required therefore to identify a true heterochromatin (or late replication) marker that might be challenging. We also wish to point out that due to the current pandemic situation of COVID-19, the laboratory at the New York University (School of Medicine) – where these studies are performed – is hardly available. We cannot define any clear schedule for performing additional dSTORM experiments, and this situation would lead to an unpredictable and disproportionate delay in the publication process.

Reviewer #2:This manuscript describes a promising new low-resolution method for the visualization and mapping of uracils in DNA. It uses UNG inhibitor UGI in combination with a TYMS inhibitor Raltitrexed (RTX) or 5FdUR to increase uracils in the genome and then uses a mutant UNG to pull-down and sequence the DNA fragments. The authors claim that their data suggest that the increased uracils are predominantly in euchromatin/early replicating regions based on epigenetic markers. While the method has several attractive features including an ability to visualize uracil foci (especially Figure 7), there are several serious shortcomings of the manuscript.1) The use of HCT116 cell line and UGI are troubling. The lack of mismatch repair (MMR) in HCT116 may affect their results. It is possible that the lack of active MMR during early replication affects the removal of uracils and thymines misincorporated across guanines. While UGI binds UNG and inhibits its action, it is unknown whether it has other physiological effects in the cells. In the very least, the results of this study should be compared with those in which MMR+ derivatives of HCT116 cells are used and UNG KO cells lines are compared with their WT parents.

The HCT116 cell line has been widely used as a model cell line in colon cancer research over the course of almost four decades (PMID: 7214343, 6956756, 22955616), for which numerous published results – including epigenomic data – are available e.g. in the ENCODE database. Colon cancer is frequently associated with deficiency in mismatch repair (MMR) (PMID: 30959407, 30442708, 28548127), and thymidylate synthase inhibitors represent a possible option as first-line treatment in the clinical practice (PMID: 24732946). Based on these data, we propose that the choice of HCT116 cell line was relevant in our study for revealing how TS inhibitors may affect uracil distribution patterns. We agree that the potential contribution of the mismatch repair to the mechanism of the effect of these drugs is a highly intriguing question that might have impact also on clinical prognosis and/or the choice of treatments. Initially, we felt that such an analysis is beyond the focus of this manuscript, but prompted by the request of the reviewer, we decided to include comparative studies on the MMR proficient version of HCT116 cells (created by the reintroduction of chromosome 3 (PMID: 8044777)), as well. For this, 12 new U-DNA-Seq experiments were performed on MMR proficient samples (shown in modified Figure 2—figure supplement 3, Figure 3, Figure 3—figure supplement 3, and the new Figure 3—figure supplement 4), we have carried out the corresponding correlation analysis (shown in modified Figure 4, Figure 4—figure supplement 2, and the new Figure 4—figure supplement 3), as well as measurement of uracil-content by dot blot (shown in new Figure 1—figure supplement 1 panel D) were also performed. Further experiments suggested by reviewer #3, namely cell cycle analysis (shown in new Figure 5) and γH2AX staining in flow cytometry (shown in new Figure 5—figure supplement 1) were also done on both the MMR deficient and MMR proficient samples. We found that MMR status does not have any observable impact on the U-DNA pattern of the non-treated samples, while in case of drug-treated ones, some drug-dependent differences were detected. We have described and analyzed these new data at different sections of the Results, as well as in the figure legends (all changes are tracked in the revised manuscript). To discuss these results, the following texts are inserted to the Discussion of the revised manuscript: “We chose the HCT116 cancer cell line that is deficient in mismatch repair…”; “It has to be noted that MMR proficiency leads to a major decrease in the correlation with early replication timing…”; “This is further supported by the fact that in MMR proficient drug-treated samples higher U-DNA content…”; “Moreover, the MMR status has markedly different influence on the resulting U-DNA pattern in case of the two drugs (cf. Figure 3-4).”; “Similarly, we also detected slightly altered cell cycle distribution patterns in case of the two drug treatments, which were differently influenced by the MMR status…”.

Regarding the use of UGI, we have to point out that UGI is a well characterized specific, small (10 kDa) proteinaceous UNG-inhibitor (PMID: 7671300). UGI expression in HEK293 cells led elevated genomic uracil content, while did not affect the cytotoxicity, the cell cycle arrest, the γH2AX signaling upon RTX or 5FdUR treatments (PMID: 17942376). We refer to these data in the Introduction of the revised manuscript (“It has already been shown that UGI expression does not affect…”). In agreement with the literature, we also detected (i) elevated U-DNA content upon drug treatments (cf. Figure 1—figure supplement 1B-D), (ii) no major physiological effect on the non-treated cells in respect to their cell cycling (cf. Figure 5A performed upon the request of the reviewer #3), and (iii) no difference in the genomic uracil distribution as compared to the non-treated wild type cells (cf. Figure 3, Figure 3—figure supplement 3, Figure 3—figure supplement 4, Figure 4, Figure 4—figure supplement 2, Figure 4—figure supplement 3). Based on these arguments, we feel that repeating all of the experiments (overall 28 sequencing, the corresponding dot blots, flow cytometry, super-resolution imaging) on UNG-KO cells would not add fundamental new insights to this present study, but may form the basis for further research.

2) Their uracil mapping software lacks a proper negative control. They calculate log2 ratio and plot this ratio across the genome for treated and untreated cells. The input DNA has not undergone the pull-down and hence the DNA pulled down using anti-FLAG antibodies will always show a different genomic distribution than the input DNA. The U-DNA sequencing results from treated cells should be normalized with respect to U-DNA sequencing reads from the pull-down of untreated samples, not the input DNA.

The reviewer suggested to use the non-treated IP samples as controls for the treated ones. In a way, we agree, and we basically did this, when we compared the log2 curves (cf. Figure 3, Figure 3—figure supplement 3), assessed their colocalization with other factors or genetic features (Figure 4, Figure 4—figure supplement 2, Figure 4—figure supplement 3), and then compared the results of the drug-treated and non-treated samples in many respects. However, for the calculation of log2 ratio and for the peak calling, we decided to use the corresponding sonicated input samples as controls. We claim that this choice was reasonable because of the following arguments.

First, sonicated genomic DNA samples provide a more balanced coverage of the reference genome, but it is still not without some fluctuation. Such fluctuation might be slightly different in case of non-treated and drug-treated cells, and therefore, it is the best to compare each IP sample (enriched in U-DNA) to their corresponding input (sonicated) ones. As expected, we did not experience highly enriched sharp peaks, therefore, such fluctuations might have a distorting impact and need to be avoided.

Second, using input as a background control for peak calling or log2 ratio enrichment calculation meets the current ENCODE standards for ChIP-seq. Currently, corresponding input is required, while previous standard suggested either input or Ig control experiment (https://www.encodeproject.org/chipseq/histone/#restrictions). Accordingly, we have inserted the following statement to the Supplementary file 1: “…as it is also recommended by the current ENCODE standard…”.

Third, given that the U-DNA patterns of the drug-treated and non-treated samples are almost complementary to each other (cf. Figure 3A and Supplementary file 2), applying non-treated samples as controls would result in a signal increase artifact rather than better description of the true enrichment.

Fourth, using the non-treated IP sample as control in the peak calling or the log2 enrichment analysis would evoke technical problems about the scaling and the handling of non-covered regions. (1) If a region is not covered by reads in case of the non-treated IP sample, then it is hard to calculate the log2 ratio. The two usual solutions for such problem may be skipping databins of zero that would result in loosing data or introducing a pseudocount that might influence the results especially at the regions of lower enrichment. (2) It is not trivial how to compare a relatively big amount of DNA immunoprecipitated from the drug-treated samples to a much smaller amount of DNA originating from the non-treated cells (5:1, cf. Figure 1—figure supplement 2A), given that the deepness of the sequencing is approximately the same for both samples.

In addition to the discussion above, we also performed blank IP experiments (using empty beads without ∆UNG sensor) and sequencing to make sure that the U-DNA patterns are valid, and not due to a specific pull-down, even in case of the non-treated samples. Data are presented in Figure 1—figure supplement 2, and Supplementary file 1—table 1. The main text was also modified accordingly, see text segment: “Still, genome-wide sequencing data could be obtained from these negative control samples as well. We demonstrated that subtracting such control signals (for details see Supplementary file 1) will not affect the detected uracil distribution pattern regardless if the sample was drug-treated or not (Figure 1—figure supplement 2B-C). These control experiments provided confidence about the applicability and specificity of our U-DNA-IP method.”. Corresponding description was added to the Materials and methods (a paragraph with the title: “Controls of U-DNA-IP method” and “High-throughput DNA sequencing and data analysis”).

3) This method is unable to pinpoint uracils in the DNA that is pulled down. This is because, the uracils are in U:A pairs and hence they cannot be distinguished from normal T:A pairs after pull-down and sequencing. This is in contrast with most of the other available methods that can map uracils to a specific base pair. For this reason, this is a low resolution alternative to existing uracil mapping methods.

We agree that our method is a low-resolution technique, but addressing the case at issue, it is relevant, adequate and robust. In the given model system, under thymidylate synthesis inhibition potentially leading to a perturbation of dUTP/dTTP ratios, uracil incorporation into the genome during either replicative or repair DNA synthesis is a basically stochastic process. Under such circumstances, characteristic U-DNA patterns might be associated with uneven genome-wide distribution of the DNA synthesis foci (e.g. at replication forks or different repair foci) and/or actual repair activities at certain regions. We claim that our approach can reliably describe these patterns. We also feel that to pinpoint the individual uracil residues would not provide additional value to this description, but could be rather misleading, if we consider that from about 10 millions of cells (=20 millions of genomes), we read on average one per million only. This issue is covered in the Discussion of the revised manuscript (see text starting with: *“*However, this aspect has lower impact, if we consider the basically stochastic nature of uracil…”).

We have reviewed the publications of other methods for determination of uracils in DNA and found that only the pre-digestion Excision-seq was explicitly stated to be capable to detect uracils with singlebase resolution. Excision-seq was used in the literature in samples with higher uracil content and smaller genome-size (PMID: 25015380). Although both UPD-seq (PMID: 31431505) and dU-seq (PMID: 29785056) use biotin-streptavidin system for the pull-down, and the biotin labeling could potentially allow site-specific detection, single-base resolution data have not been shown in either of these publications. In the revised manuscript, the first paragraph of the Discussion has been rephrased to provide a more elaborate comparison of the available methods.

The U-DNA-Seq method we present includes a refined analysis pipeline to ascertain its status as a useful alternative of the existing U-DNA sequencing methods. The use of the catalytically inactive UNG-based sensor renders it possible to complement the genome-wide distribution patterns with cellular localization studies. This advantage is also better underlined in the revised version, following the suggestion of the reviewer #3.

4) The manuscript lacks a proper positive control. It may be useful for the authors to introduce uracils by other means such introduction of AID gene to introduce uracils in the Ig locus and then demonstrate that their method can detect such localization of uracils. In the absence of a positive control, it is hard to know whether their method works correctly. It is possible- for example- that the treatment of cells with RTX or 5FdUR arrests the cells in early S phase and that is why the uracil incorporation occurs in early replicating regions. In which case, the lack of uracil-enriched fragments from the late replicating heterochromatin regions is a somewhat trivial result.

We agree that a relevant positive control is valuable. Reviewer #2 suggested expression of AID to introduce uracils specifically into the Ig locus in order to detect such localization of uracils by our method. However, overexpression of AID without its proper regulation and targeting is expected to lead to deamination events outside the Ig loci also, as it was reported in case of tumorigenesis (PMID: 26845615). We therefore designed another positive control experiment. Namely, a uracil-containing spike-in sequence was added to genomic DNA fragments, then the U-DNA-IP was performed and the enrichment of the spike-in was followed by qPCR, as it has been already mentioned above in the reply to the Editor’s main point #2. Corresponding sentences are inserted to the Results (“To further confirm the capability of U-DNA-IP, uracil-containing spike-in DNA was combined with non-treated genomic DNA samples (cf. Materials and methods). In these samples U-DNA-IP led to 4.5 fold enrichment of the uracil-containing spike-in DNA compared to the uracil-free spike-in as determined by qPCR.”), and to the Materials and methods (a paragraph with the Title: “Controls of U-DNA-IP method”).

We note that the robustness and the reproducibility over biological replicates reported in our study also support the applicability of our method. Moreover, comparison of U-DNA-Seq results to the independent dU-seq data also confirmed the reliability of our method (cf. Discussion of the submitted manuscript (*“*Re-analysis of the earlier published dU-seq data…*”*) and Appendix 1).

The reviewer also mentioned the possibility “that the treatment of cells with RTX or 5FdUR arrests the cells in early S phase and that is why the uracil incorporation occurs in early replicating regions. In which case, the lack of uracil-enriched fragments from the late replicating heterochromatin regions is a somewhat trivial result.”. We agree that the S-phase arrest will influence the pattern of uracil incorporation, and now in the revised manuscript, it is presented and discussed more extensively. Clear correlation has already been shown between the U-DNA pattern and early replicating segments in the submitted manuscript (cf. Figure 4C-D). Further genome segmentation analysis, presented in the revised manuscript, also confirmed this (Figure 4—figure supplement 3). In addition, upon the request of the reviewer #3, cell cycle analysis was also performed (Figure 5) supporting the presence of a cell cycle arrest upon the applied drug treatment. The connection of the cell cycle arrest and the correlation between U-DNA pattern and early replicating segments is now presented in the revised manuscript (cf. Figure 4C-D and Figure 5). We propose that these results provide additional insights. U-DNA-Seq might give a molecular approach to the replication arrest, while flow cytometry provides a phenotypic description of cell cycle phases. Interestingly, drug dependent and also MMR status dependent differences were detected by both approaches, which are discussed in the Discussion of the revised manuscript (“Similarly, we also detected slightly altered cell cycle distribution patterns…”). It also has to be noted that correlation with the replication timing is not exclusive, and DNA synthesis coupled to either transcriptional activity or epigenetic remodeling might also have an impact on the resulting genomic U-DNA pattern in the drug-treated samples.

5) The authors dismiss the dU-seq (Shu et al., 2018) rather lightly. Shu and colleagues verified their findings by enrichment of centromeric regions showed by LC-MS/MS that centromeric uracils increased in UNG-knockout cells. Finally, they reintroduced UNG in the cells and showed that this reversed the uracil localization to centromeres. As noted above, the authors of the current manuscript have not done such careful controls in their experiments.

It is our intention to provide a useful comparison among the relevant methods. Towards this end, we present a well detailed assessment of the dU-seq method in the Discussion section, and also in Appendix 1. We do not question the validity and reliability of the LC-MS/MS data of the Shu et al. publication and to emphasize this, we added the following sentence into the Discussion section: *“*It has to be noted that the authors also applied additional experimental approaches, like 3D-PCR for sensitive detection of U:G pairs, mass spectrometry on genomic DNA fraction enriched in centromeric DNA regions.”

However, there are serious concerns about the capability of dU-seq method to reveal centromeric localization (cf. Appendix 1). These are the following: (1) Short-reads were mapped to the highly repetitive centromeres using a routine mapping software without any extra care or quality control. (2) The data analysis pipeline is not published in sufficient clear details, major points are missing, e.g. whether only the uniquely mapped reads were used or not, whether blacklist and deduplication were applied or not, or which reference genome set was used. (3) Their peak calling was also not reported correctly, it is not fully described if they used controls, and which parameters they set in the MACS2. (4) Their probable approach was to subtract peaks of the control experiment from the peaks of pulled down samples, which is also questionable (cf. above). (5) Their results are shared on the GEO only partially, making a critical evaluation quite problematic.

Finally, we agree with the reviewer that our method has the novel potential for direct visualization of uracils in human genomic DNA. Here, we demonstrated that our technique allows (1) detection of genomic uracil in eukaryotic cells, (2) co-staining with other factors, and (3) measurement of co-localization for the predicted correlating factors from the U-DNA-Seq results. These are important steps towards its application in more sophisticated experimental setup also combined with time-resolved U-DNA-Seq and RNA-seq that might lead to deeper understanding of the molecular mechanisms, and especially the impact of uracil incorporation upon treatments with TS inhibitors. Our present results contribute novel information relevant to existing questions such as: (1) What kind of genomic uracil patterns can arise upon two widely used thymidylate synthase inhibitor drugs in a valid cancer cell line model? (2) Is there any correlation between the U-DNA patterns characteristic for these drug treatments and epigenetic factors or other genomic features? (3) Is there any connection between the known cell cycle arrest and the U-DNA pattern? (4) Is there any difference between the U-DNA patterns induced by treatments with the two drugs? (5) Is there any impact of the MMR status on the U-DNA pattern and does it correlate with mechanistic differences? Further studies based on our present results will reveal more mechanistic details.

Reviewer #3:The authors develop a new approach to measure the location and density of incorporated uracils in the human genome. They identify an interesting pattern of uracil incorporation in the human genome under normal and thymidine-depleted (using TS inhibitors) conditions, and associate these patterns with specific chromatin features. They also generalize the tool to visualize uracil incorporation with fluorescence microscopy. Both of these are significant advances of use to the DNA repair community. Moreover, these data may eventually be used to probe molecular and cellular consequences caused by common chemotherapies.1) This study would be improved by employing computational methods designed to identify patterns across multiple independent genomic signals. The one-by-one correlations the authors present provide some insight into general correlations between uracil density, replication timing, and histone modifications, but this ad hoc approach could miss important correlations across all these signals. The segway software (Hoffman, Noble, et al) was developed specifically for this purpose and could be applied to data they have in hand to understand broader genome-wide correlations.

We highly appreciate this suggestion and fully agree. In the revised version, we have performed the suggested Segway analysis on the U-DNA-Seq data also including those obtained from MMR proficient cells (latter experiments were performed according to the request of reviewer #2). We have used ChIP-seq data on HCT116 cells available in the ENCODE database, and also used data from our own H3K36me3 ChIP-seq experiment (performed according to the request of reviewer #1). The results shed light on the relations of the different correlating factors and features, suggesting that none of the histone markers alone can explain the detected U-DNA patterns, and that the correlation with the replication timing or the chromatin structure should be taken into account with higher impact. The resulted signal distribution is presented in Figure 4B and further calculations regarding the replication timing scores and AT content of these genomic segments are shown in Figure 4—figure supplement 3. Corresponding texts are inserted into the respective figure legends; the Materials and methods (“Genome segmentation analysis on our U-DNA-Seq data,…", and “Replication timing scores and AT content were calculated on the genomic segments…”); the Results (a complete section starting with “To understand broader genome-wide correlations, a genome segmentation approach was employed…“, and two sentences “The replication timing correlation and the AT content were also calculated…”, and “Furthermore, Segway analysis strengthened…”); and the Discussion (“Such combinatorial behavior was further demonstrated by the genome segmentation analysis…"). The applied commands/scripts are reported in the Supplementary file 3 with the subtitle “Genome segmentation analysis of U-DNA-Seq results and ChIP-seq data from the ENCODE using Segway genome segmentation algorithm.”. The title of Supplementary file 3 was changed accordingly. Similarly, Supplementary file 4 was completed with the calculation of replication timing and AT content.

2) More characterization of any DNA damage response in these cells would be helpful to understand whether human uracil incorporation patterns are a "neutral" pattern in the absence of DDR (e.g., similar to high levels of seemingly innocuous uracil incorporation found in dut ung *E. coli* and yeast) , or represent a stress-induced condition caused by uracil incorporation followed by some checkpoint activation. Some flow cytometry would be useful to see the effect of RTX on cycling of these cells, and H2AX phosphorylation could be measured in bulk by western and in the microscopy experiments upon treatment.

We agree that more characterization on the phenotype caused by the drug treatments could improve our understanding of the U-DNA pattern. Hence, based on the suggestion of the reviewer, we performed two types of flow cytometry experiments addressing the potential effect of drug treatments on the cell cycle progression as well as on the DNA-damage signaling under the same conditions that were used in the U-DNA-Seq and the colocalization dSTORM ICC experiments.

First, we addressed the cell cycle using propidium iodide and 5′-bromo-2′-deoxyuridine (BrdU) staining in all those HCT116 samples, for which we provided U-DNA-Seq data. Our results can be summarized as follows. (1) In non-treated cells, UGI expression did not cause any visible changes in the cell cycle progression. (2) Both RTX and 5FdUR treatment (in the presence of UGI) caused significant changes, resulting in arrest-like patterns, which are not identical in case of the two drugs, in good agreement with the detected differences of the corresponding U-DNA patterns. (3) MMR status has an influence on these patterns, which is in good agreement with the MMR status related changes in the correlations between U-DNA patterns and replication timing. These results are presented in Figure 5 of the revised manuscript. Corresponding texts are inserted into the figure legend; the Results (a paragraph that is started with *“*As the uracil distribution pattern in drug-treated cells shows correlation…”); the Discussion (the following sentences: “We demonstrated that UGI expression did not cause any observable…”, “… or the less tight control on cell cycle arrest (cf. Figure 5) allowing more extended replicative synthesis.”, “Similarly, we also detected slightly altered cell cycle distribution patterns in case of the two drug treatments…”), and the Materials and methods (a section with the title “Cell cycle analysis and γH2AX staining”).

Second, regarding the potentially induced DNA damage response (DDR), we also performed flow cytometry measurements using γH2AX immunostaining. The results revealed an increased γH2AX level indicative for DDR in case of both drug treatments. These results are shown in the Figure 5—figure supplement 1. Corresponding texts are inserted into the figure legend; the Results (“As expected (cf. (Meyers et al., 2001)), 5FdUR and RTX treatment…”); the Discussion (“However, equally induced DNA damage response (reported by γH2AX)…”); and the Materials and methods (“Occurrence of DSBs was investigated by immunofluorescent staining of γH2AX…”).

3) Given previous indications that replication timing is somehow linked to uracil incorporation, more discussion is warranted to reconcile similarities and differences in this relationship in *E. coli*, yeast, and human uracil incorporation patterns relative to replication timing and associated changes in nucleotide pools.

To discuss the correlation between uracil incorporation and replication timing, we have inserted a new paragraph into the Discussion starting with “Our data showing that under normal conditions…”. Here, we show differences between the timing of genomic uracil incorporation in our drug-treated cells and in other biological systems previously published in the literature, and the possible reasons are also discussed.

[Editors’ note: what follows is the authors’ response to the second round of review.]

Revisions:The following minor revisions are strongly recommended for acceptance to publish in eLife.1) The criticism of other methods are given the disproportionate significance in the discussion. Although the effort to highlight the advance and innovation of the technique used in the current manuscript over other previously published methods is completely valid and helpful, overly long and highly critical discussion of other methods diminishes the meaningful discussion of the findings in this paper and becomes unnecessary distraction. There is also a risk of misstating what the other methods do or point out deficiencies that may have simple explanations. A separate review paper comparing various methods would better suit such discussion. I recommend that the authors reorganize the Discussion section so that it starts by describing what this manuscript has accomplished and briefly summarize the pros and cons of all the methods towards the end.

We agree with the Editors’ recommendation, and accordingly we have restructured the Discussion. Comparison of the different methods was significantly shortened and moved from the beginning of the Discussion (see paragraph starting with "As we demonstrated here, the genome-wide uracil distribution patterns have relevance…"). For harmonization, a new first paragraph is inserted in the Discussion starting with "Here we focus on the alteration of U-DNA distribution pattern".

2) Clarification is required regarding the replication scores derived from Weddington et al. Is the distribution of scores correlated with timing (negative = early, positive = late?). It's unfortunate that ENCODE Repli-seq data is not available for the HCT116 cell line; this would have been a better comparison.

Thank you for pointing out the meaning of replication score was not defined. We have now inserted the following sentences with an extra reference into the figure legends of Figure 4—figure supplement 2: "Replication timing scores are derived from E/L Repli-seq experiments, where cycling cells are pulse-labeled with BrdU, then sorted to early and late S-phase fractions by flow cytometry, and BrdU labeled genomic DNA fragments are pulled down and subjected to NGS. Signal tracks are computed from the read coverages in early over late S-phase samples, therefore the higher score means earlier replication (Marchal et al., 2018). The scale goes from -2.5 to 5." Similar description is inserted to the figure legend of Figure 4—figure supplement 3: "Replication timing scores are derived from E/L Repli-seq experiments, the higher score means earlier replication (Marchal et al., 2018)."

Please add discussion or speculate on the possible mechanism regarding why uracil enrichment upon treatment both appear at similar replication timing, relative to the control samples (Figure 4—figure supplement 3)?

This is already discussed in details in the following sections:

"Uracil appearance *via* thymine replacing misincorporation implies prior DNA synthesis involved in either replication, or transcription-coupled DNA repair, or epigenetic reprogramming (e. g. erasing the methyl-cytosine epigenetic mark). Importantly, we found that uracil pattern showed the highest correlation with the features (early replication, active promoters and DNase hypersensitive sites, and CpG islands) linked exactly to these processes (cf. Figure 4C)."

With even more details starting with: "The antifolate or nucleotide-based thymidylate synthase inhibitors, such as 5-FU, RTX or 5FdUR are known to lead to cell cycle arrest…"

We also inserted a short reference: "also supported by Figure 4—figure supplement 3".